# Non-invasive MR imaging of human brain lymphatic networks with connections to cervical lymph nodes

Mehmet Sait Albayram ◑ [1✉], Garrett Smith ◑ [1], Fatih Tufan ◑ [2], Ibrahim Sacit Tuna ◑ [1], Mehmet Bostancıklıoğlu[3], Michael Zile ◑ [4,5] & Onder Albayram ◑ [4,6,7]

Meningeal lymphatic vessels have been described in animal studies, but limited comparable data is available in human studies. Here we show dural lymphatic structures along the dural venous sinuses in dorsal regions and along cranial nerves in the ventral regions in the human brain. 3D T2-Fluid Attenuated Inversion Recovery magnetic resonance imaging relies on internal signals of protein rich lymphatic fluid rather than contrast media and is used in the present study to visualize the major human dural lymphatic structures. Moreover we detect direct connections between lymphatic fluid channels along the cranial nerves and vascular structures and the cervical lymph nodes. We also identify age-related cervical lymph node atrophy and thickening of lymphatics channels in both dorsal and ventral regions, findings which reflect the reduced lymphatic output of the aged brain.

[1] Department of Radiology, University of Florida, College of Medicine, Gainesville, FL 32610, USA. [2] Geriatrician (PP), Silivrikapi Mh. Hisaralti Cd, Istanbul 34093, Turkey. [3] Elysium Health Center, Gaziantep 27090, Turkey. [4] Division of Cardiology, Department of Medicine, Medical University of South Carolina, Charleston, SC 29425, USA. [5] Division of Cardiology, Department of Medicine, Ralph H. Johnson Department of Veterans Affairs Medical Center, Charleston, SC 29425, USA. [6] Department of Neuroscience, Medical University of South Carolina, Charleston, SC 29425, USA. [7] Hollings Cancer Center, Medical University of South Carolina, Charleston, SC 29425, USA. ✉email: albaym@radiology.ufl.edu

Macromolecules, waste products, and excess fluid from most tissues are known to drain into the systemic lymphatic system[1–5]. It is not well defined how waste materials or interstitial fluid (ISF) in brain parenchyma are cleared. These materials or fluids are conventionally understood to enter the cerebrospinal fluid (CSF) then drain to the systemic circulation through the arachnoid granulations[6,7]. Older animal studies have demonstrated CSF drainage via arachnoid granulation into the venous blood. More recent animal studies have demonstrated CSF-ISF drainage via meningeal lymphatic vessels (MLVs) and along the cranial nerves (CNs) into deep cervical lymph nodes (dcLNs)[1–5,8–12]. No convincing in vivo imaging to date has shown CSF egress through arachnoid granulations in the human brain. Human studies also show similar evidence of CSF-ISF drainage through MLVs especially along the parasagittal dura, but evidence of similar drainage along CNs in humans is scarce and previously documented along the olfactory nerves through the cribriform plate in the autopsy studies[2,13,14]. Recently, glymphatic drainage from the live human brain to cervical lymph nodes has been shown by MR imaging[15]. In spite of animal studies, the exact contribution of other cranial nerves to CSF-ISF drainage is not known in live human[9,11–13].

Paolo Mascagni in 1787 is regarded as the first to describe dura mater lymphatics. Subsequent studies have observed these structures through use of electron microscopy, histology and immunohistochemistry[16–18]. Two studies recently re-demonstrated with modern techniques the existence of MLVs in the brains of rats[1,2]. Subsequent studies described similar lymphatic tissues in the human brain. MLVs described in these human studies were visualized with magnetic resonance (MR) imaging[19–24] and with confocal microscopy[25]. A non-contrast MR study identified flow within parasagittal MLVs in the opposite direction of blood in the adjacent superior sagittal sinus using MR angiography technique[24]. Animal studies have described networks of MLVs within the dorsal and ventral cranial meninges[1,2,26,27], with the ventral system being the dominant. In human studies, however, CSF-ISF drainage has been documented almost exclusively within the MVLs around the dural sinuses-parasagittal space in the dorsal compartment and MVLs of anterior cranial fossa along the olfactory nerves[19–22,25]. Indirect depiction of CSF-ISF flow from dorsal MVLs to dcLNs has been achieved in humans in a few studies[10,15,21]. While animal studies have confirmed that CSF-ISF drainage occurs predominantly through a ventral drainage system, human studies have depicted drainage primarily from a dorsal system rather than a ventral system. In humans, the full course of connections between CSF-ISF flow from the MVLs to cervical LNs are not known. A more complete visualization of MLVs and their connections with LNs in humans may be possible using more modern imaging technologies.

Disruption of glymphatic and MVLs functions likely underlies or contributes to many clinical conditions such as traumatic brain injury, Alzheimer disease, multiple sclerosis[8,25,28–34]. Differences in glymphatic and MVL morphology and function by sex and age may also be related to sex and age differences in incidence of certain neurological diseases. Recently, imparied meningeal lymphatic drainage in patients with idiopathic Parkinson's disease has been described with contrast enhanced MR[35]. While MRI can allow for noninvasive in vivo evaluation of MLVs for these pathologies[29], this has previously been achieved only through use of intravenous and intrathecal Gadolinium (Gd)[19–22]. Gd use is not completely safe for healthy subjects or patients due to Gd retention in the body, and visualization of MLVs using this approach is limited to anatomic compartments physically accessible to Gd, as described above[36]. Therefore, additional noninvasive MR techniques are needed that do not rely on Gd to allow safer, more complete visualization of MVLs for further studies relating to MVLs or CSF-ISF drainage pathways.

Here, we show a noninvasive, non-contrast 3D fluid attenuated inversion recovery (FLAIR) MR method permitting detailed visualization of dorsal-along the venous sinuses and ventral mLVs- around perineural/peridural spaces of cranial nerves, CSF-ISF drainage around nerves in the human brain, as well as visualization of direct relationships among these pathways and dcLNs. Age and sex differences in mLVs is also reported.

## Results

**MR phantom study.** MR technique is very important for optimization of signals at specific protein concentrations. In our study, increasing TE values resulted in elevated relative SI across the protein samples analyzed, with the highest protein-agar-ratio observed at TE of 601 ms (Fig. 1a, b). Mild signal increase relative to H2O can be seen from 10 to 20 mg/dl, but a decrease in signal back toward that of H2O is observed at concentrations of 40–160 mg/dl levels. Further increase in protein concentration from 320 to 1280 mg/dl results in a marked increase in signal (Fig. 1b). Increasing TI values also resulted in increasing relative SI across the protein samples analyzed, with the higher protein-agar-ratio observed at TI of 1600 ms (Fig. 1c, d). Additionally, signal decrease back toward that of H2O occurred at concentrations of 60 mg/dl levels at TI of 1400 ms and at 240 mg/dl levels at TI of 1600 ms. Drastic signal changes were therefore observed across the protein solutions at different MR parameters in our phantom studies. We assume proper MR parameter selection is critical to specific visualization imaging of brain lymphatics. Further improved signal-noise-ratio of brain lymphatics may be achievable with additional MR parameter manipulation in future studies.

We found that signal within expected regions of dural lymphatics corresponded to protein concentration ranging from 2000 to 4000 mg/dl with 3D-T2-FLAIR sequence parameters set at TR/TE/TI = 5000/386/1400 msec, suggesting that regions of high protein and macromolecular concentration are present in lymphatic structures along the sagittal sinuses. This feature allows visualization of lymphatic tissues using our technique (Fig. 1e–h; Supplementary Table 1). Indeed, our 3D-T2-FLAIR sequence parameters (TR/TE/TI = 5000/386/1800 msec) yielded no significant signal within regions of CSF, where protein concentration is 15–60 mg/dl (Fig. 1f). This phenomenon reflects signal suppression observed in CSF spaces with 3D T2-FLAIR images. Our measurements of protein concentration most likely do not reflect precise in-vivo concentrations due to differences in temperature and molecular constitution between the human body and phantom. Lymphatic fluid includes not only albumin, but also cells, cell fragments, lipids and other solutes. Therefore, the behaviour of the natural brain lymphatics fluid should be different from the albumin solutions in this study. Albumin concentration in interstitial fluid is very similar to its concentration in lymph (~1.6–1.8 g/dl or 1600–1800 mg/dl)[37,38], which is consistent with findings in our phantom study. Our measurement is higher than the systemic lymphatic fluid protein concentration. This difference might be related to the different components of the lymphatic fluid such as cells, solutes, debris and other proteins. These results, however, provide validation for our technique in visualization of lymphatic structures without use of contrast, providing strong signals within regions of lymphatic tissue and lymph nodes, while minimizing signal from vascular structures, white matter, muscle, bone and soft tissue including fat (Fig. 1h, Supplementary Table 2).

**Human subjects.** A total of 81 human subjects (45 females and 36 males) with a mean age of 41.7 (SD 20.4, range 15–80) years were

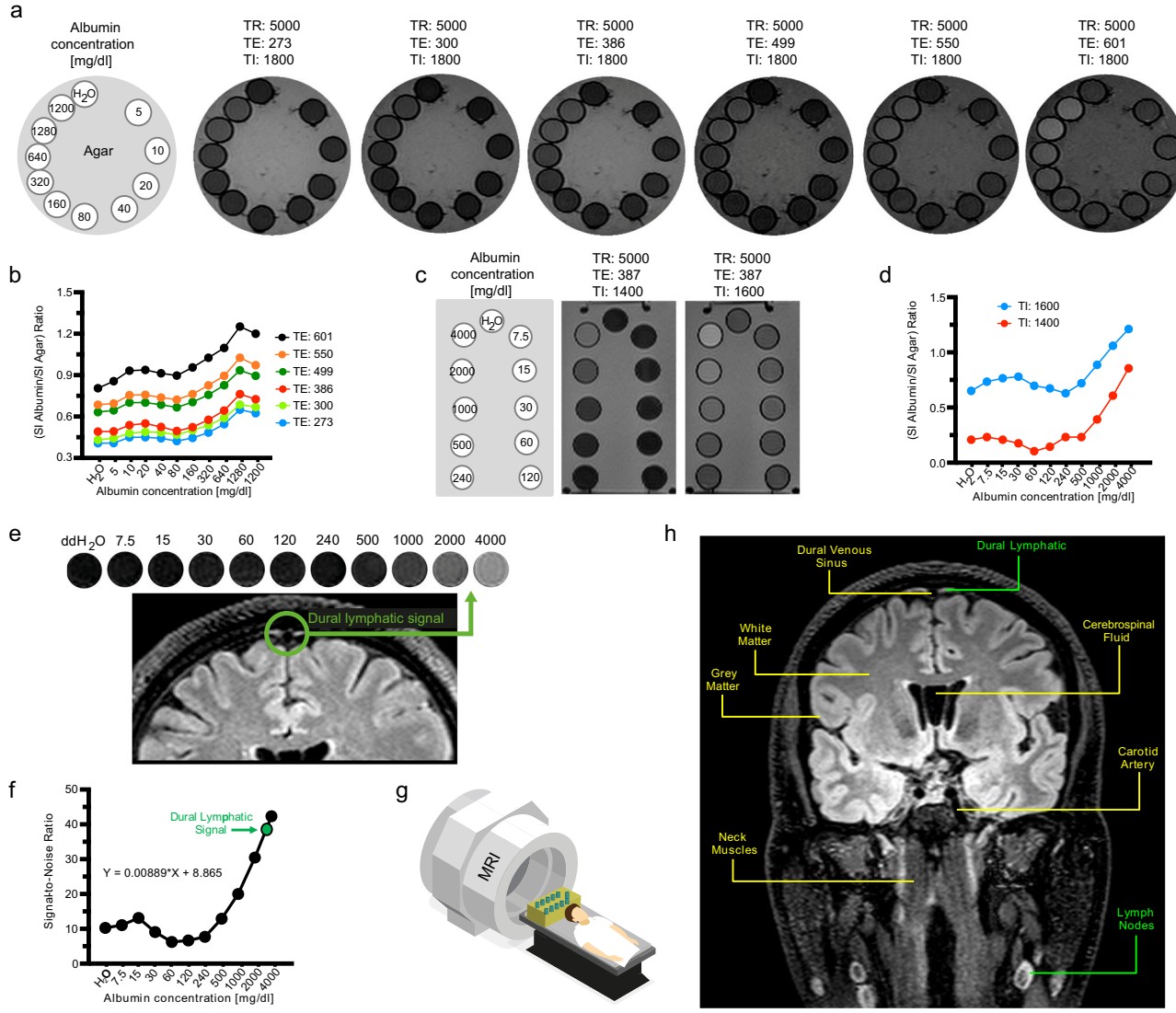

**Fig. 1 Phantom studies of head and neck structures using 3D-T2 FLAIR.** Echo time (TE) related changes in 3D-T2 FLAIR signal can be seen in six phantom images (**a**), with the strongest signals observed at TE 601 in solutions of higher protein concentrations. The ratio of albumin signal intensity (SI) to agar SI with respect to albumin concentration can be seen in (**b**). Increasing TE values resulted in increasing signal-noise-ratio (SNR) across the protein samples analyzed, with the highest SNR observed at TE of 601 ms. The rectangular phantom images depict inversion time (TI) related signal changes related to albumin concentration (**c**). Signal suppression of water was more prominent at TI 1400 ms, and signal was significantly greater at all protein concentrations at TI 1600 ms. The ratio of SI albumin to SI agar with respect to albumin concentration can be seen in (**d**) for IR 1400 ms and 1600 ms. The greater TI value of 1600 ms yielded a greater SNR across the protein samples analyzed. Simultaneous MR imaging was performed of a rectangular phantom and a healthy adult male subject (**e**–**g**). Parasagittal dural lymphatic signal intensity corresponds to that of the higher albumin concentrations between 2000 and 4000 mg/dl (**e**). SNR shows a positive relationship with albumin concentration up to 15 mg/dl, then a negative relationship from 15 to 60 mg/dl, followed by a marked increase in SNR between 60 and 4000 mg/dl (**f**). An example image identifying key structures in the head and neck region from which signal intensity measurements were obtained, including the cervical lymph nodes and parasagittal dural lymphatics, is provided in (**h**). These findings demonstrate that this T2-FLAIR sequence's parameters are sensitive to lymphatic fluid and lymphatic tissue without the need for contrast agents. TR relaxation time.

included in this retrospective study (Fig. 2). This study was approved by the local institutional review board (IRB201902528) and conducted with data from the Picture Archiving and Communication Systems (PACS/ Visage 7.1.14, Visage Imaging®). Since no new patient information was collected for the study, and all data in PACS are de-identified, the IRB approved the study as a retrospective review with a waiver for patient consent. M.S.A collected data from MR images and I.S.T performed post-processing of certain images. None of the authors were involved in direct interactions with the patients or in the direct collection of images/data from the patients.

**Intra-rater reliability**. The intra-class correlation coefficient values for intra-rater reliability ranged between 0.983 and 1 for SI values and between 0.995 and 1 for thickness values ($p < 0.001$ for all). The intra-rater variability was excellent at all regions tested for these ten cases.

**Volume measurements**. The mean volumes of GM, WM, TB, CSF, and IC were 689.1 cm$^3$ (SD 93.5), 522 cm$^3$ (SD 106.3), 1211 cm$^3$ (SD 166), 177.7 cm$^3$ (SD 84.5), and 1388.8 cm$^3$ (SD 160.4), respectively.

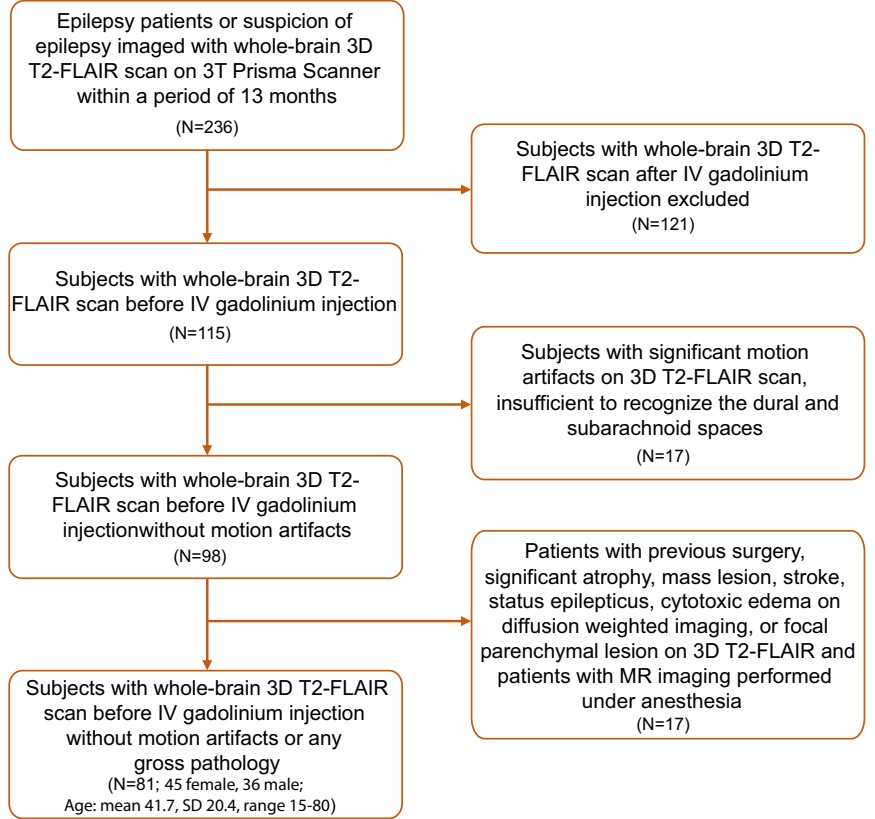

**Fig. 2 Study profile describing the patient selection process.** In this figure, application of our inclusion and exclusion criteria to the study population is represented in detail. SD, standard deviation.

**Visibility, description and comparison of the dorsal and ventral lymphatic systems.** Anterior and middle aspects of the sagittal sinus lymphatic drainage system were visible in all subjects while the posterior aspect was visible in 59 of 81 subjects. Sinus rectus, confluence, left sigmoid sinus-transverse sinus junction, and posterior foramen magnum were visible in all of the subjects while other dorsal regions were not visible in all subjects. The transverse sinus (57/81), right sigmoid sinus-transverse sinus junction (80/81), right sigmoid sinus (63/81), left sigmoid sinus (61/81), left jugular vein (47/81), and right jugular vein (32/81) were visible in some of the subjects. All components of the ventral system were visible in all subjects. Some lymphatic structures may have been too small to visualize with this technique due to the resolution of 0.9 mm slice thickness used. The position of the head may have also impaired visibility of posteriorly located meningeal dorsal structures in some patients due to compression of these structures by brain parenchyma due to gravity.

Structural attributes of the dorsal and ventral mLVs are shown respectively in Figs. 3–4 and Supplementary Tables 3–4. SI and thickness values of the dorsal and ventral systems are listed, respectively, in Figs. 3–4 and Supplementary Tables 3–4. The overall median thicknesses of the dorsal and ventral lymphatic systems were positively correlated (r = 0.45, p < 0.001; age, sex, and ICV adjusted partial correlation r = 0.32, p = 0.04). The overall mean SIs of the dorsal and ventral lymphatic systems were positively correlated (r = 0.49, p < 0.001; age, sex, and intracranial volume adjusted partial correlation r = 0.55, p < 0.001). The CSF volume was positively correlated with overall mean dorsal thickness (r = 0.67, p < 0.001), overall mean ventral thickness (r = 0.39, p < 0.001), and overall mean dorsal SI/WM (White matter) SI (r = 0.47, p < 0.001), but not with overall mean ventral SI/WM SI (r = 0.01, p = 0.96). Gray matter, white matter, or total

parenchymal (gray + white matter) volumes were not correlated with overall mean dorsal or ventral SI/Muscle SI or thickness values.

The overall median thickness of the dorsal system (2.23 [IQR 0.7]) was greater than that of the ventral system (1.95 [IQR 0.5], p < 0.001). The overall mean SI of the ventral system/WM SI (2.05 [SD 0.26]) was greater than that of the dorsal system (1.07 [SD 0.17], p < 0.001).

Internal carotid arteries, Jugular veins, cranial nerve complex IX–XII in the vascular space, and connections between lymph nodes in the neck region.

The visibility of connections between the walls of the ICA, jugular veins, and CN complex in the vascular space and vascular lymph nodes are shown in Fig. 5 and Supplementary Table 5. Connections between the ICAs, jugular veins and CNs IX–XII in the vascular space are seen in all patients. Connection of these structures with cervical LNs, including retropharyngeal LNs and vascular LNs were visible in almost all subjects (Fig. 5 and Supplementary Table 5). Additional figures are provided in Supplementary Figs. 1–3. All of the connections were visible in the younger subjects but not in the older ones. Additionally, the SI ratio of the CN IX–XII complex to central white matter (CWM) inferior to the cervical lymph nodes (LNs) was significantly lower than that of the same ratio superior to the cervical LNs (1.40 [SD 0.21] vs. 0.44 [SD 0.08], p < 0.001). These findings show direct and indirect evidence of the connection of the meningeal lymphatics and deep cervical lymph nodes in living humans.

*Aging-associated changes in brain lymphatic system.* The age of 50 years discriminated well between lower and higher than mean dorsal (area under the curve 0.68, 95% CI 0.56–0.79, p = 0.007,

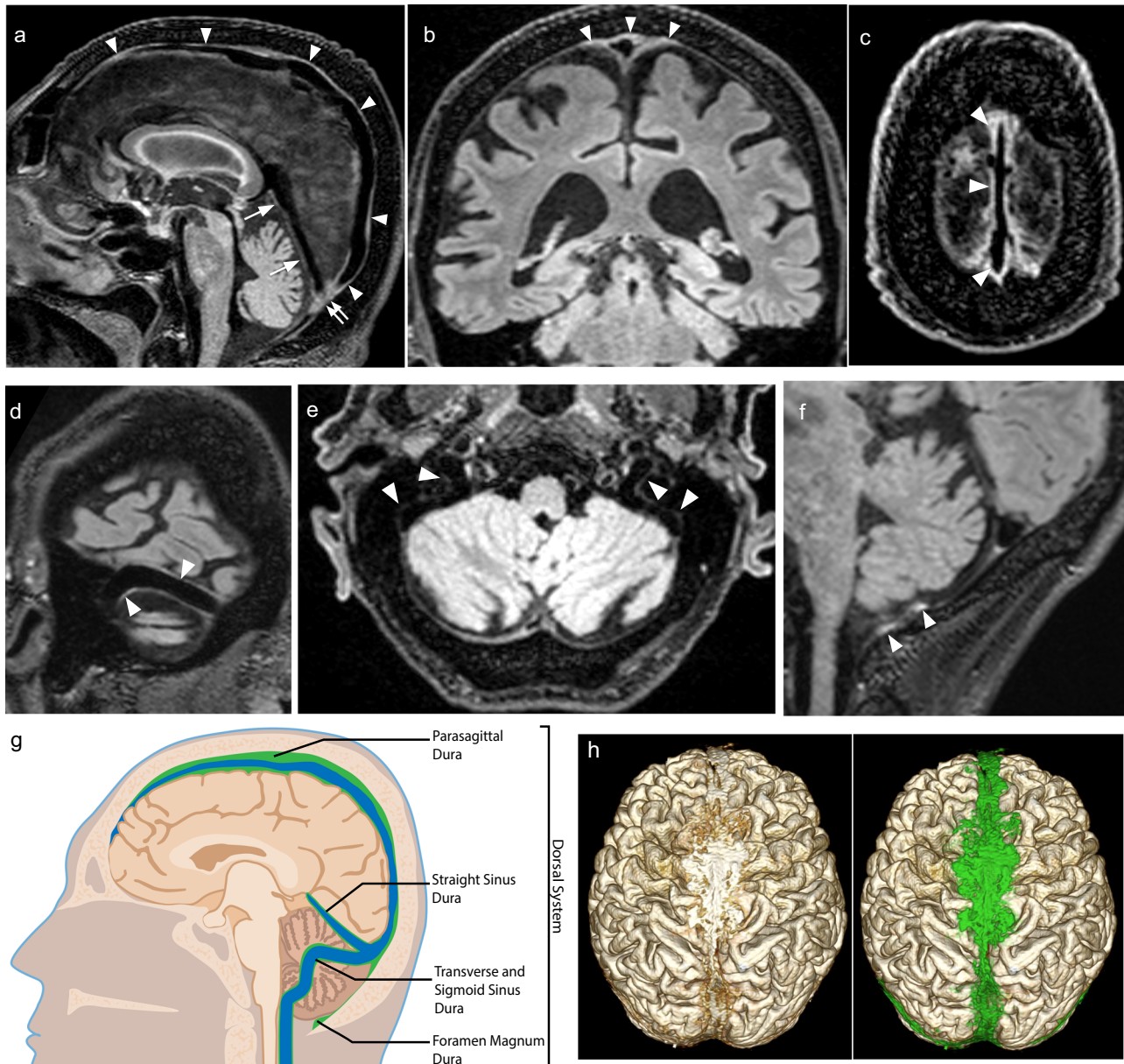

**Fig. 3 Dorsal dural lymphatic system MR images and metrics from multiple subjects. a** Sagittal FLAIR image shows linear signal increase representing the dural-parasagittal lymphatics around the walls of the superior sagittal sinus (arrowheads), straight sinus (arrows) and confluent sinus (double arrow). Coronal (**b**) and axial (**c**) FLAIR images depict thick dural-parasagittal lymphatic signals along the sagittal sinus (arrowheads) and the cortical veins. Similar but less prominent lymphatic signals are also seen along the wall, the transverse sinus (**d**, arrowheads), sigmoid sinus, and jugular vein (**e**, arrowheads). **f** Sagittal image depicts the dural lymphatic structures in the posterior aspect of the foramen magnum (arrowheads). **g** Illustration of distribution and relative volume of dorsal dural lymphatic structures (green) and venous system (blue) derived from T2-FLAIR. **h**. 3D representation of dural lymphatics as defined from T2-FLAIR shown from dorsal view. The dural lymphatics (bright parasagittal irregular structures) are defined from T2-FLAIR and co-registered with rough segmentation of the brain and the dural-parasagittal lymphatics (green).

sensitivity 50%, and specificity 76.3%) and ventral SI/thickness ratios (area under the curve 0.73, 95% CI 0.62–0.85, $p < 0.001$, sensitivity 58%, and specificity 85.4%). In binary regression analyses, an age older than 50 was associated with lower dorsal and ventral SI/thickness ratios, independent of sex and total intracranial volume (brain + CSF) measurement (OR 2.9, 95% CI 1.1–7.7, $p = 0.029$ and OR 8.3, 95% CI 2.8–24.7, $p < 0.001$, respectively). There were 29 subjects in the older group (>50 years old) and 52 subjects in the younger group. The older and younger age groups had similar sex ratios (females 55.2% and 55.8%, respectively, $p = 0.96$). While the older group had lower GM and WM volumes (647.9 cm$^3$ [SD 68.7] vs. 712 cm$^3$ [SD 98.1],

$p = 0.001$ and 484.6 cm$^3$ [SD 91] vs. 542.8 cm$^3$ [SD 109.3], $p = 0.017$, respectively), they had higher CSF volumes (252.5 cm$^3$ [83.9] vs. 136.1 cm$^3$ [SD 48.9], $p < 0.001$) compared with the younger group. Older subjects had higher SI ratio and thickness values in most aspects of the dorsal system, but lower SI ratio and higher thickness values in the ventral system compared with younger ones (Fig. 6; Supplementary Table 6). The overall mean SI ratio of the ventral system was significantly higher than that of the dorsal system in both younger (259.9 [SD 32.6] vs. 130.4 [SD 23.6], $p < 0.001$) and older subjects (247.4 [SD 35.4] vs. 140.3 [SD 21.1], $p < 0.001$) (Fig. 6a, b, d, e). The overall median thickness of the dorsal system was greater than that of the ventral system in

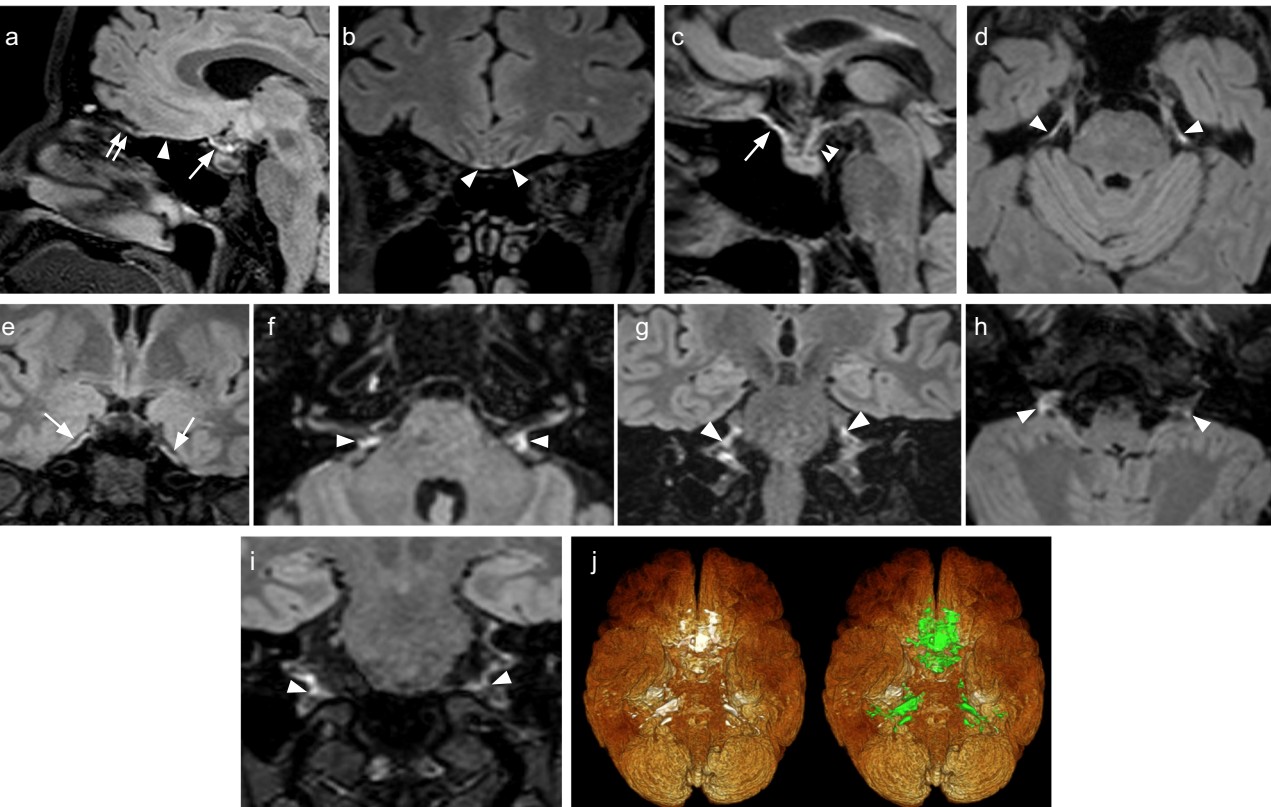

**Fig. 4 Ventral dural lymphatic system MR images and metrics from multiple subjects. a–c** FLAIR images show the ventral dural lymphatic elements in the anterior cranial fossa (arrowheads), olfactory bulb (double arrow), optic grove (arrows) and diaphragma sella (double arrowhead; **c**). **d, e** Ventral dural lymphatic elements can be seen at the level of the orifices of neural foramina of the trigeminal nerves (**d**, arrowheads) and edge of Meckel's cave dura (arrows; **e**). Ventral dural lymphatic elements can also be seen at the level of the orifices of the internal auditory canals on axial (**f**, arrowheads) and coronal (**g**, arrowheads) views. These structures are visible as well at the level of the jugular foramina around the IX–XI complex (arrowheads) on axial (**h**) and coronal (**i**) and views (arrowheads). **j** 3D representation of ventral dural lymphatics as defined from T2-FLAIR shown from ventral view. The ventral dural lymphatics (bright ventral irregular structures around the ventral dural surfaces and neural foramina) as defined from T2-FLAIR and co-registered with rough segmentation of the brain and the dural lymphatics (green).

both younger (2.1 [IQR 0.5] vs. 1.9 [IQR 0.5], $p = 0.036$) and older subjects (2.6 [IQR 0.8] vs. 2 [IQR 0.5], $p < 0.001$) (Fig. 6 a, b, g, h). The dorsal/ventral SI ratio and dorsal/ventral thickness ratio were higher in the older subjects compared with younger ones (Fig. 6f, i). The average SI/thickness ratios of the dorsal and ventral systems were lower in older subjects compared with younger ones (Fig. 6j, k). The older group had significantly lower SI and short axis measurements in retropharyngeal and vascular LNs regions (Fig. 6l). Age related changes in the rate of visibility and comparisons in this context can be seen in the Fig. 6c for the dorsal system. Posterior sagittal sinus, transverse sinus, sigmoid sinus, and jugular vein were more commonly visible in the older group. The ventral elements were visible in all cases of the older and younger age groups.

There was a positive correlation between age and CSF volume ($r = 0.75$, $p < 0.001$) and negative correlations between age and GM and WM volumes ($r = -0.48$, $p < 0.001$ and $r = -0.34$, $p = 0.002$, respectively). Age was positively correlated with mean dorsal thickness and SI/WM SI ($r = 0.69$, $p < 0.001$ and $r = 0.43$, $p < 0.001$, respectively) and mean ventral thickness ($r = 0.26$, $p = 0.024$), but not with ventral SI/WM SI ($r = -0.13$, $p = 0.25$), when adjusted for sex and total intracranial volume (white + gray matter + CSF).

*Sex-associated changes in brain lymphatic system.* The comparison of male and female subjects is summarized in Supplementary Table 7. Male subjects had higher GM, WM, and CSF volumes

compared with female subjects (726.8 cm³ [SD 91.2] vs. 658.9 cm³ [SD 84.8], $p = 0.001$; 562.1 cm³ [SD 112.9] vs. 489.9 cm³ [SD 89.6], $p = 0.002$; and 201.6 cm³ [SD 94.9] vs. 158.7 cm³ [SD 70.6], $p = 0.022$, respectively). The dorsal SI and ventral SI as well as dorsal/ventral SI ratio were higher in males than females (Fig. 7a–e). Although the overall median thickness of the dorsal and ventral systems were higher in males than females (Fig. 7f, g), median dorsal/ventral thickness ratio and mean SI/thickness ratios of the dorsal and ventral systems were similar between male and female subjects (Fig. 7h–j). Male and female subjects had similar SI and short axis measurements in retropharyngeal and vascular LN regions (Fig. 7k).

Sex was correlated with average dorsal thickness ($r = -0.43$, $p < 0.001$) and SI ($r = -0.41$, $p < 0.001$); average ventral thickness ($r = -0.33$, $p = 0.001$); and ventral SI/thickness ratio ($r = 0.23$, $p = 0.04$), but not with average ventral SI or dorsal SI/thickness ratio when adjustment for age and total intracranial volume was made. In binary multivariate regression analyses, sex was not independently associated with a high/low dorsal or ventral SI/thickness ratio (covariates: age and total intracranial volume).

## Discussion

Our study provides a detailed visualization of both dorsal and ventral brain lymphatics in live humans acquired by MR imaging without the use of contrast media. Dorsal MLVs were apparent around almost all dural venous-parasagittal structures in our study and were continuous with fluid channels from the skull

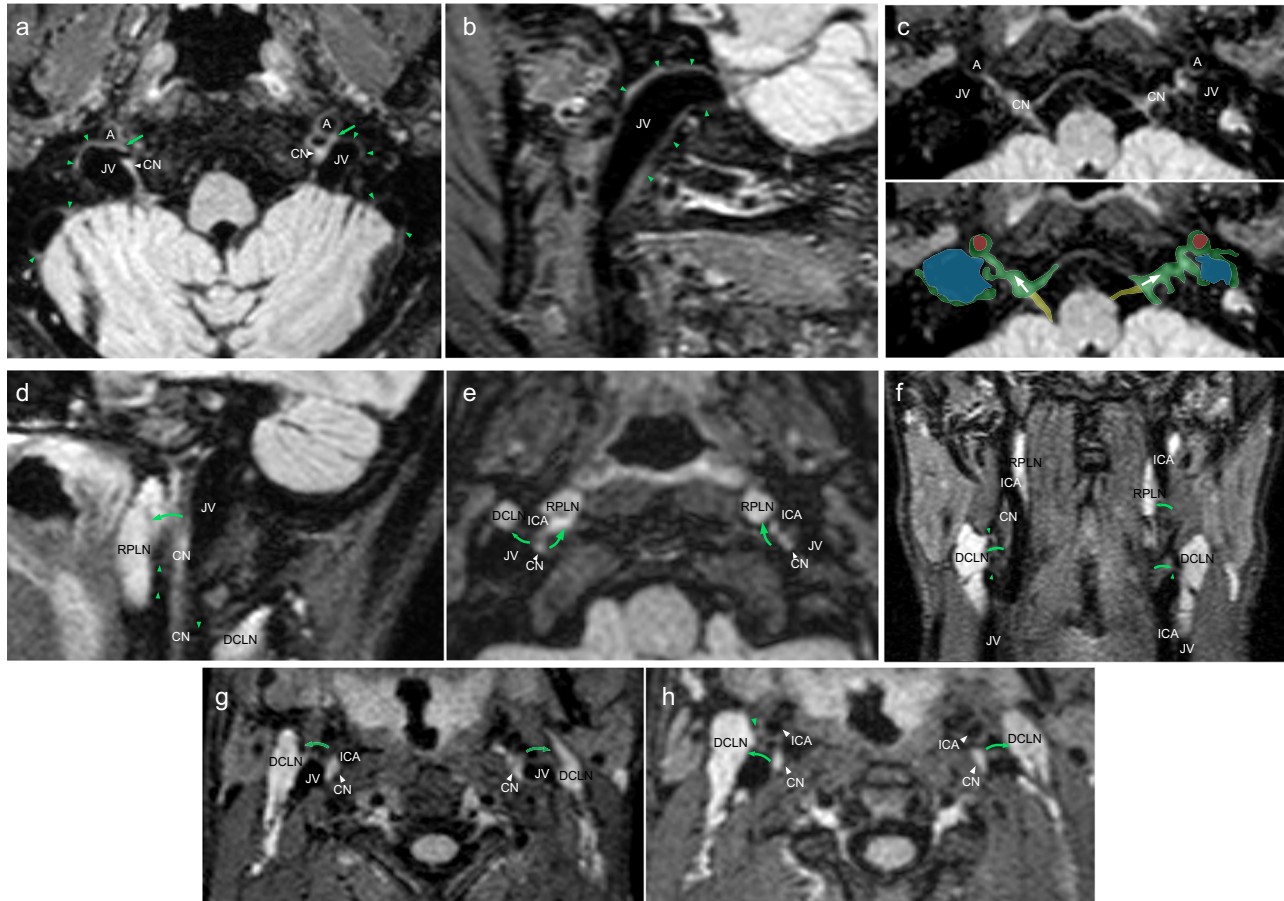

**Fig. 5 Depictions of lymphatic drainage at the levels of the skull bases and neck.** Axial (**a**) and sagittal (**b**) FLAIR images show prominent lymphatic fluid signals along the wall of the jugular vein (JV) in the upper neck region (green arrowheads). **a** Lymphatic signal connections with internal carotid artery (A), JV and cranial nerves (CN) can be seen (green arrows). **c** Axial magnified image reveals lymphatic signal communications along the cranial nerve IX–XI complex (CN), internal carotid artery (A) and jugular vein (JV). Corresponding color overlay depicts veins (blue), meningeal lymphatic tissue/flow (green; arrows), ICAs (red), and CNs (yellow). Sagittal (**d**), axial (**e, g, h**), and coronal (**f**) FLAIR images depict lymphatic connections from the cranial nerve IX–XI complex to the deep cervical lymph nodes (DCLN; green arrows and green arrowheads) and retropharyngeal lymph nodes (RPLN; green arrows and green arrowheads). Similar types of connections can also be seen between CNs and the ICAs as well as between the ICAs and lymph nodes (green arrows and green arrowheads).

base to the dcLNs. In previous studies, these structures were seen only around the sagittal sinus and transverse sinus, and their connections between each other and skull base were not clearly identified[19–24]. Ventral dural lymphatics were also not previously well described in humans, with MLVs identified only in the anterior cranial fossa[19,23]. In our study, however, the ventral meningeal lymphatics system can be seen easily at the level of cranial foramina and around the cranial nerves. Both dorsal and ventral systems are similarly prominent in this study, in contrast to findings from animal studies.

The previously described primary route of CSF-ISF absorption to MVLs was along the superior sagittal sinus in human studies, contrasting with findings from mice studies which concluded that CSF-ISF outflow occurred predominantly along cranial nerves in the ventral region[20,26]. We detected evidence of CSF-ISF drainage channels in both dorsal and ventral cranial compartments. The dorsal system included channels adjacent to the venous sinuses, jugular veins and within the posterior aspect of the foramen magnum, while the ventral system was most prominent along the CNs at the level of the neural foramina. The average signal across all ventral structures was higher than that of the dorsal structures in this study, in which all scans were conducted in a supine position. In rodents, whose telencephalon is relatively small, a

ventral drainage system may be sufficient to manage all CSF-ISF waste outflow. In humans, however, an additional or exaggerated dorsal CSF-ISF drainage system may be necessary to manage all CSF-ISF waste produced in a significantly larger telencephalon. The weak lymphatic signal and discontinuity in occipital region along the posterior aspect of the sagittal sinus in the human brain may be due to focal compression of the occipital lobe against the meningeal lymphatic structures by posterior movement of the brain itself due to gravity while in the supine position. It is unknown, however, how head posture affects CSF and brain lymphatic drainage or the relative engagement of the dorsal and ventral systems in humans[39,40]. We scanned human subjects in the supine position, and, possibly due to the effect of gravity, signals from the posterior sagittal sinus, transverse sinus, and sigmoid sinus-transverse sinus are less prominent than those of the mid sagittal sinus. Rodent studies are generally done in the prone position, freeing the superiorly located dorsal system from the effects of gravity. In a recent rodent study by Ahn and colleagues[26], significant CSF efflux was observed to occur via basal outflow area-Jugular foramen, whereas negligible CSF efflux was observed via dorsal regions along the superior sagittal sinus in the prone position. Additionally, Lee and colleagues[39] identified an effect of body posture on CSF efflux pathways in humans,

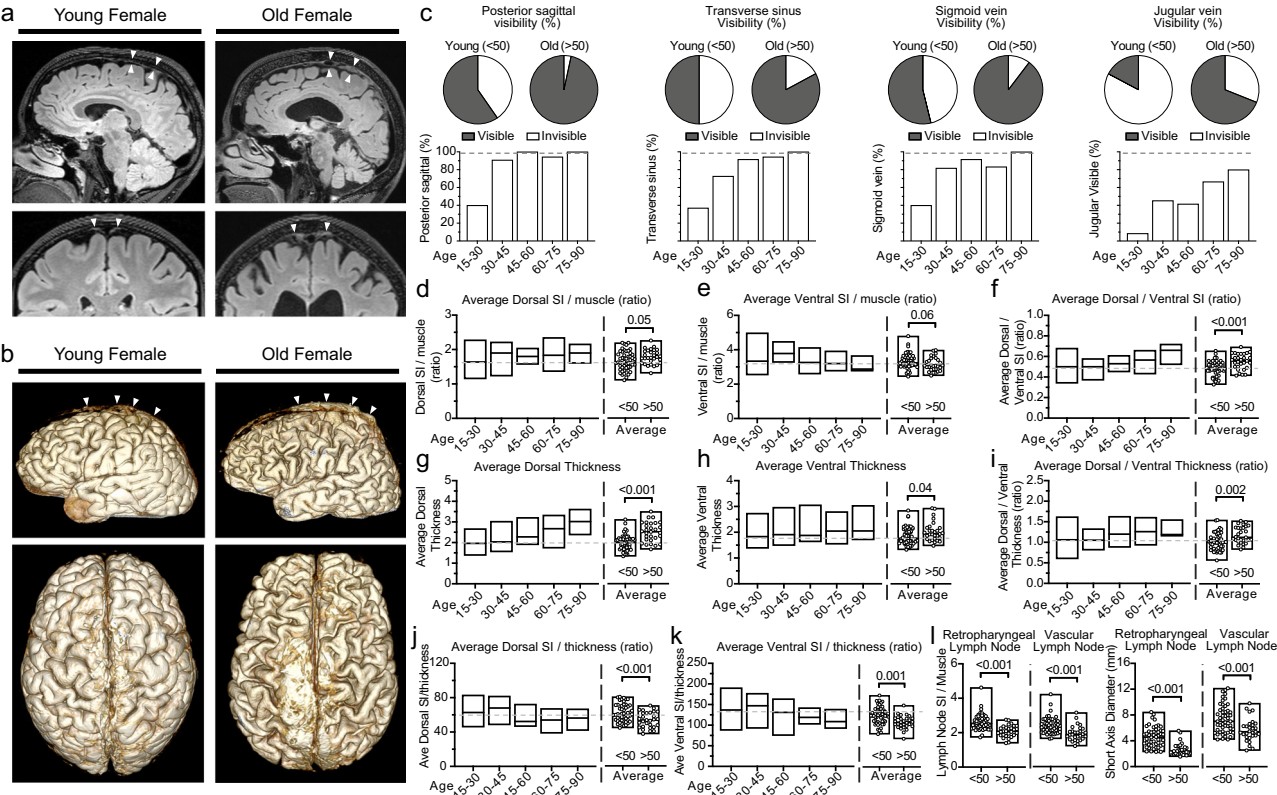

**Fig. 6 Age effects on dorsal and ventral dural lymphatic structures. a** Dorsal parasagittal dural lymphatic system in young and aged female subjects. Sagittal FLAIR images and coronal magnified FLAIR images reveal a significant increase in thickness and extension of the parasagittal dural lymphatics structures along the sagittal sinus (arrowheads) in the aged subject. **b** Comparisons of the 3D representation of dorsal parasagittal dural lymphatics as defined from T2-FLAIR. Images depict the brain for the lateral (upper image) and dorsal (lower image) views. **c** Comparison of visibility of dorsal lymphatics between older and younger age groups and across decades of age. **d–k** Comparison of thickness and signal intensity values of dorsal and ventral lymphatic systems between older and younger age groups and across the age groups. Student $t$ test was used to compare older and younger age groups. All tests are two-sided. **l** Comparisons of retropharyngeal and vascular lymph node SI and short diameter by age groups. Student $t$ test was used to compare older and younger age groups. All tests are two-sided. Data are presented as mean values ± SEM. [Age: 15–30; $N = 35$], [Age: 30–45; $N = 10$], [Age: 45–60; $N = 16$], [Age: 60–75; $N = 15$], [Age: 75–90; $N = 5$] and [Age: <50; $N = 52$] and [Age: >50; $N = 29$].

describing reduced efflux along the superior sagittal sinus in the supine position compared to the prone position. Efflux from VIII nerve & cochlear regions was not significantly different between prone and supine positions[39]. We suspect the dorsal and ventral lymphatic drainage balance or ratio of CSF-ISF in humans might be different in the prone, supine, upright, right-left lateral positions.

The brain must be able to expel both aqueous and macromolecular waste products. Classically, absorption of CSF occurred through arachnoid granulations and villi of the intracranial and spinal venous sinuses[6]. Several studies have established that some CSF-ISF drains to dcLNs via the cranial nerves or meningeal lymphatics[1–5,8–10,26,41,42]. More recently, it has been suggested that MLVs in the dura drain fluid, solutes, and cells from the brain parenchyma to dcLNs but the direct anatomical pathways for such drainage have not been demonstrated[43]. Our data also support the dual absorption hypothesis that solute rich ISF or lymphatic fluid drains into the dural lymphatic system and then into dcLNs[5,6,43]. If both the aqueous and macromolecular components of CSF were completely absorbed along either the arachnoid granulation system or dural lymphatic pathway, we would not expect to see evidence of increased macromolecular concentration, measured by signal increase, in MVLs seen in our study. CSF signal, such as within the ventricular system, is hypointense on FLAIR imaging, but we observed hyperintense signals in the lymphatic networks along the dorsal veins and CNs,

which can be explained by increased macromolecular concentration in specific regions optimized for MVLs absorption. These findings support the model of dual absorption of CSF, with arachnoid granulations specialized for absorption of aqueous CSF components and dural lymphatics specialized for macromolecular clearance[5,6,43].

Recent studies suggest that CSF drainage may occur along two additional pathways: MLVs and perineural pathways[8,9,30,31,34,44]. In the literature, they are generally accepted as different systems[8,31,34], but it is possible that they intersect at the level of the skull base. Currently, it is unknown whether ISF waste from the brain is transported in channels surrounding cranial nerves or within the nerves themselves[8]. It is not clear how lymphatic flow from MLVs is channeled through the skull base to dcLNs given that the intracranial dura end at the skull base[8]. The epineurium of the cranial nerves and spinal nerves is continuous with the dura mater[45–47]. Some studies have suggested that the proximal portions of extracranial CN segments possess lymphatic conduits in the epineurium[48]. We observed significant signatures of lymphatic flow surrounding the CNs at the neural foramina and extracranial skull base. SI of lymphatic flow around CNs within the neck was significantly decreased inferior to dcLNs, and we identified direct communications between CNs and LNs in almost all subjects, further suggesting possible diversion of lymphatic flow from nerves to lymph nodes (Fig. 5). Our observation suggests that ISF and CSF drain from the intracranial meningeal

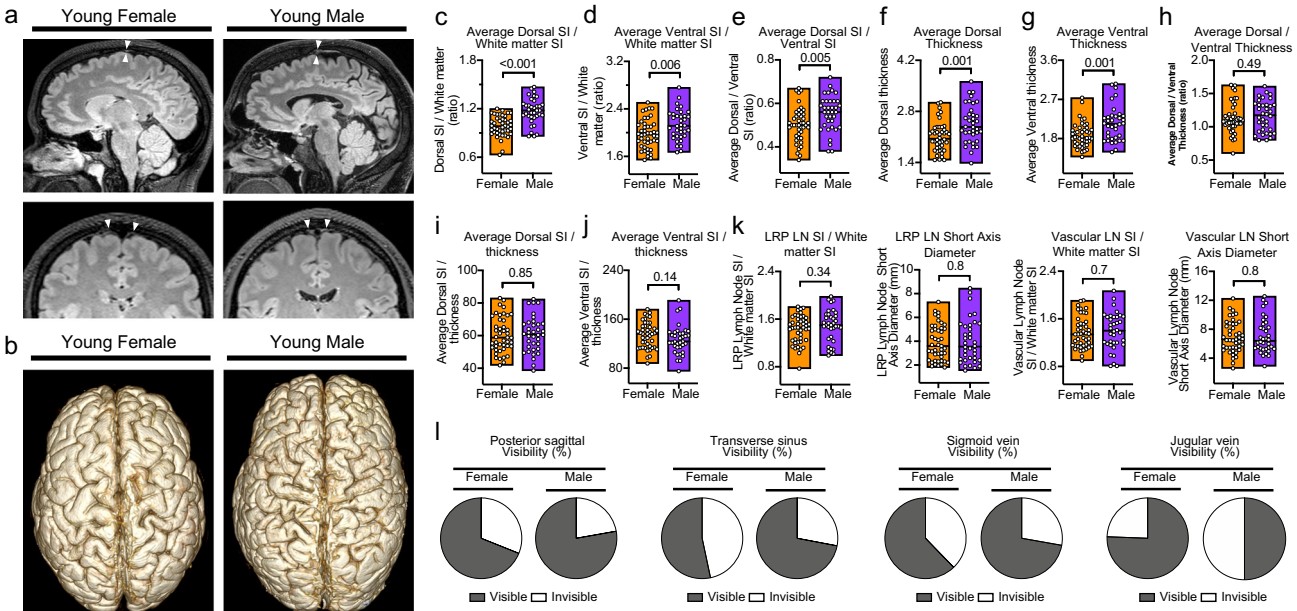

**Fig. 7 Comparisons of the dorsal and ventral dural lymphatic system by sex. a** Representative sagittal FLAIR images and coronal magnified FLAIR images reveal increased thickness, extension, and SI of the parasagittal; dural lymphatics structures along the sagittal sinus (arrowheads) in the male subject compared to the female subject (both the male and female are young adults of the same age). **b** Comparisons of the 3D representation of dorsal dural lymphatics as defined from T2-FLAIR shown from dorsal view. **c–j** Comparison of thickness and signal intensity values of dorsal and ventral lymphatic systems between male and female subjects. Student *t* test was used to compare older and younger age groups. All tests are two-sided. **k** Comparisons of retropharyngeal and vascular lymph node SI and short diameter by sex. Student *t* test was used to compare older and younger age groups. All tests are two-sided. **l** Comparison of visibility of dorsal lymphatics by sex. Data are presented as mean values ± SEM. [Female; $N = 45$], [Male; $N = 36$].

lymphatics to the cervical lymph nodes through the neural foramina at the skull base. Since the dura and epineurium are continuous, we hypothesize that the flow of CSF-ISF along the dura is continuous with the epineurium as well, and that these epineural lymphatic channels ultimately drain into the dcLNs (Fig. 8). This is consistent with observations by Asplenund and colleagues that lymphatic vessels drain from within the skull along the dura mater of cranial nerves, not in the nerve itself[1]. Lymphatic marker-positive cells in postmortem human brain samples have recently been identified in the perineurium and endoneurium of cranial nerves by immunohistochemical techniques[49]. The authors suggested that soluble waste may move from the brain parenchyma via perivascular and paravascular routes to the closest subarachnoid space and then travel along the dura mater and/or cranial nerves[49]. We also believe that meningeal lymphatics and perineural drainage pathways are not separate systems, but rather are part of the same waste management pipeline; however, direct CSF-ISF drainage in the cranial nerve itself via the endoneurium, running adjacent to nerve axons, can not be evaluated in this study. Further studies are needed for clarification.

According to our study result, the vascular-carotid space in the neck is very important for the CSF-ISF drainage from the brain. The carotid artery, jugular vein and cranial nerves are located in this space. Recently, the lymphatic vessels connecting the lymph nodes located adjacent to the internal jugular vein within the carotid sheath at the level of upper neck region have been shown in autopsy study, supporting the possibility of continuous flow along the dorsal system from superior sagittal sinus to the jugular vein and the lymph nodes in the neck[50]. We believe our findings support this theory in living humans, but that our images likely partially conceal the continuity of this signal due to supine positioning and gravity effects as discussed before. Aspelund and colleagues identified lymphatic vessels adjacent to the ICA immediately below the skull in the neck, and identified that brain

ISF drains into cervical lymph nodes via the efferent carotid lymphatic vessels in the neck[1]. Additionally, Clapham R. et al. showed a direct relationship between the cervical ICAs and dcLNs in a human autopsy study[51]. In our study, we detected direct communication between the cranial nerves and the deep lymph nodes in the neck including retropharyngeal and vascular space lymph nodes. The dorsal lymphatic flow via jugular vein wall and the ventral lymphatic flow along CNs IX–XII drains via vascular space to the deep cervical lymph nodes. The lymphatic signals along the wall of internal carotid arteries can be seen in our study with connections with lymph nodes and cranial nerve or jugular vein signals. We believe this signal might be mainly indirect flow from cranial nerves or jugular foramen and belongs to lymphatic rete in the vascular space. We observed signals in the upper cervical ICA and in the skull base, and in connections of these structures to dcLNs. We were simply unable to detect continuous flow from intracranial arteries to cervical ICA. There are two main hypotheses proposed to explain drainage of ISF-lymphatic fluid from the brain: Glymphatic Theory[52] and the Intramural peri-arterial drainage (IPAD) pathway[26,43,53–57] (iPAD) concept. In the Glymphatic Theory, solute- and macromolecule-rich ISF-CSF drains via meningeal lymphatics and eventually reaches cervical lymph nodes[52]. According to the iPAD concept, ISF mainly drains towards intracranial arteries which then drain into the cervical lymph nodes, and meningeal lymphatics drain only CSF[54]. Overall our findings suggest that drainage of CSF-ISF occurs along multiple dural lymphatic structures including multiple cranial nerves and dural venous sinuses-parasagittal sinus space, but no definite evidence of periarterial drainage can be seen in our study. We detected significant ventral lymphatic flow outlets along the anterior cranial fossa, optic groove, Meckel cave, and internal auditory canal; however, our technique is unable to depict lymphatic drainage out of certain CNs and their connections to lymphatic tissue, such as CNs I, II, V, and the VII/VIII complex. We provided a

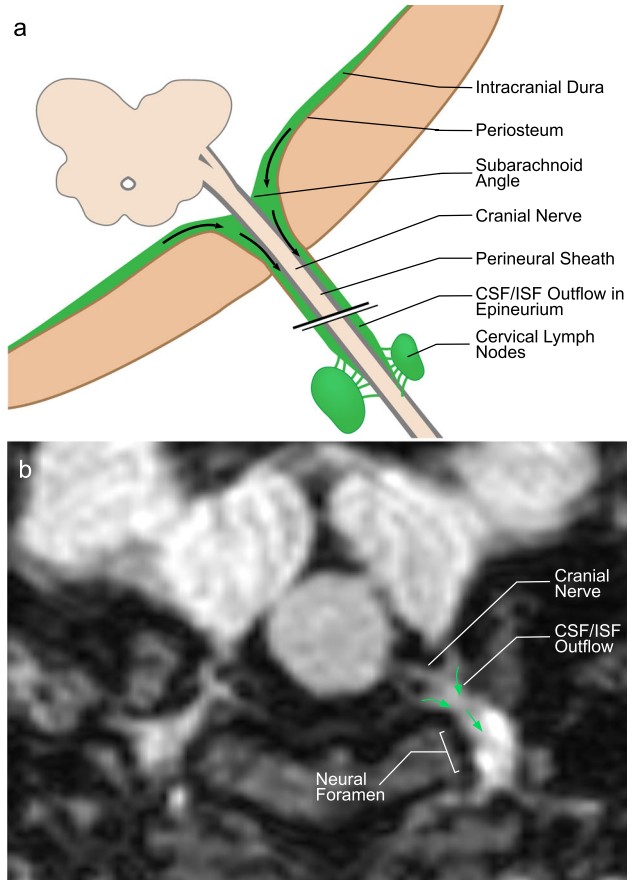

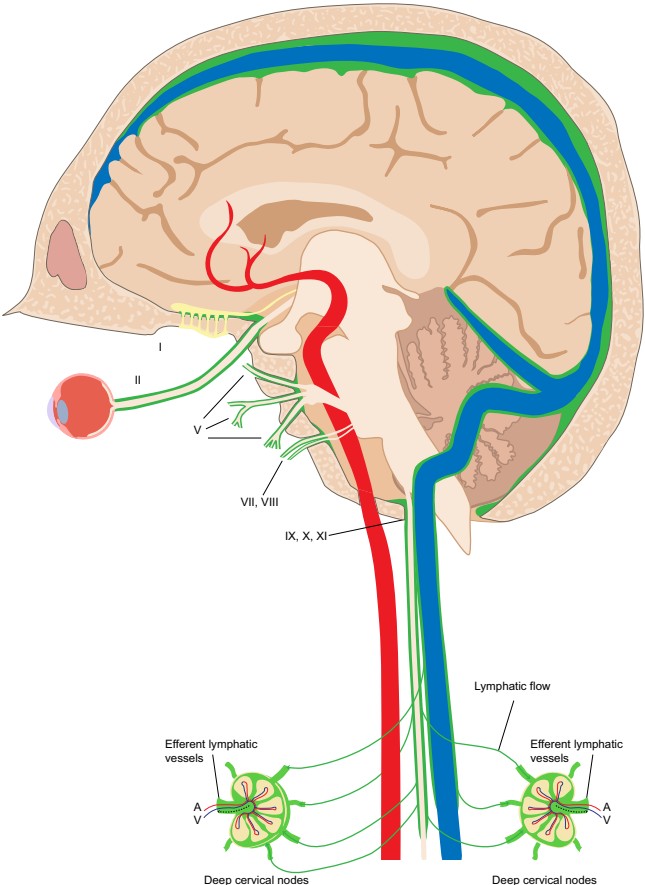

**Fig. 8 A schematic model of cranial nerve, dura, and dural lymphatic flow at cranial foramina. a** Illustration of hypothetical lymphatic flow from dural lymphatics to the epineurium of a generic extra-cranial nerve. Dural lymphatic flow in this model becomes prominent at the level of the foramen before entering the epineural space. This lymphatic flow follows the cranial nerve epineurium and later routes to cervical lymph nodes. **b** Axial MR image reflecting this model showing dural lymphatic signal extending into the neural foramen and along the cranial nerve outside of the skull (arrows indicate direction of flow).

**Fig. 9 Schematic model of the brain lymphatic system and its drainage pathways.** The overall system is divided into dorsal and ventral systems. The dorsal system includes ISF-lymphatic fluid present along dural venous sinuses that drains along the jugular veins and foramen magnum. The thickest of these fluid spaces occurs at the midsagittal level. Decreased caliber of this space present at the level of the posterior sagittal sinus is most likely due to effects of gravity while in the supine position during scanning. The ventral system includes ISF-lymphatic fluid along the olfactory nerves in the anterior cranial fossa, optic groove, dura at the orifice of the Meckel's caves, dura at the orifice of the internal auditory canals, and dura at the level of the orifices of jugular foramen. The green signal at the orifices of neural foramina represents lymphatic fluid. These pathways channel ISF-lymphatic fluid outside of the intracranial along the space epidural surfaces of the cranial nerves. ISF-Lymphatic signals along the cranial nerves, jugular veins and petrous and cervical ICAs in the neck show connections between each other and cervical lymph nodes. The schematic depicts direct connections between meningeal ISF-Lymphatic fluid to deep neck nodes via multiple neural foramina and along the skull base structures in humans.

schematic model of the brain lymphatic system and its drainage pathways based on our findings (Fig. 9).

Significant decline in CSF lymphatic outflow via meningeal lymphatic vessels has been described in aged mice compared to young[27]. This meningeal lymphatic dysfunction has been proposed as a potential exacerbator of AD pathology and age-associated cognitive decline[58]. Thus, augmentation of meningeal lymphatic function may be a promising therapeutic strategy for preventing or delaying age-associated neurological diseases[58]. Age is associated with increased branching and hyperplasia of basal mLVs as well as significant regression of dorsal mLV branches in mice[26]. Human MRI studies have also documented dural-parasagittal lymphatic dysfunction in aged subjects[21]. Recently, increased meningeal lymphatic volume along the sagittal sinus associated with advanced age was found in the brains of human cancer patients using MR imaging[59]. Aligned with these findings, we identified increased thickness of mLVs with age in the ventral systems, and even more so in the dorsal systems (Fig. 6 and Supplementary Table 6). In animals, basal mLVs have also been observed to increase in size and show highly branched and hyperplastic phenotypes in aged mice when compared with young mice[26]. Aged mice, however, have shown regression of dorsal mLV branches and diminished coverage of the SSS by

dorsal mLVs compared with young mice[26]. Our findings of prominent change within the dorsal structures in humans may be related to morphological differences between mouse and human brains such as the more developed telencephalon in humans. Body position during imaging may further contribute to the observed inter-species differences; as animal studies are generally performed in the prone position, whereas our study was done with patients in the supine position. Dorsal/ventral SI and dorsal/ventral thickness ratios were also increased with age, representing an age-related shift in tissue prominence from the ventral to dorsal system, which again differs from findings in animals. Both diameter and maximum SI of the cervical LNs (retropharyngeal and vascular) decreased with age as well. Atrophy of these LNs

may contribute to backup of lymphatic flow resulting in expansion of the dorsal and ventral systems. Additionally, age is associated with reduced muscle cell mass within the lymphatic vascular wall, which is associated with contractile dysfunction and enlarged lymphatic diameter[60]. In our study, dorsal and ventral SI/thickness ratios were also decreased with age. These findings might also reflect age-related lymphatic vessel dilatation and contractile dysfunction similar to findings made in previous animal and human studies[21,26,27]. The existence of morphological changes of meningeal lymphatics in aging raised other interesting questions that should be addressed in future studies such as meningeal lymphatic changes in the Alzheimer population.

Despite the age-related changes in the meningeal lymphatics and the sex related changes in the extracranial lymphatic vessels, sex differences in the human intracranial dorsal and ventral lymphatic systems have not been previously described in the literature. We found only one recent study that described a higher meningeal lymphatic space volume along the sagittal sinus in males compared to females[59]. In the present study, average dorsal and ventral tissue thickness and SI, as well as dorsal/ventral average SI ratio, were higher in males than females. This result shows correlation with the mentioned study. The greater tissue thickness observed in males may be due to higher overall brain volume in males compared to females. Sex hormones may affect brain lymphatics given their known role in lymphatic tissue development and remodeling[26]. Dorsal and ventral SI/thickness ratios, however, showed no sex differences, suggesting similar overall lymphatic throughput in males and females which is unique finding from our study.

Our study has several limitations. First, recognition of the mLVs was based on previous references lacking corresponding histological comparisons. Second, MR image resolution was limited to a voxel size of 0.9 mm. Third, we studied only patients with a history of seizure disorder or suspicion of seizure, albeit without significant focal lesions. We excluded all cases with parenchymal lesion, stroke, or intracranial surgery as these may alter visualization of lymphatic structures. We also excluded patients having undergone sedation or having experienced status epilepticus, as sedation and acute prolonged seizure may change glial lymphatic flow and visualization of lymphatic structures. However, our study population is not composed of healthy subjects; chronic seizure might change visualization of the lymphatic structures as well. This represents a limitation of the present study. Our phantom study, however, suggested that the meningeal lymphatics signals can be seen in a healthy subject as well; however, lymphatic fluid includes not only albumin, but also cells, cell fragments, lipids, and other solutes. We therefore anticipate behaviour of natural brain lymphatic fluid to be different from the albumin solutions as used in this study. Fourth, we used point-sized ROI measurements for SI due to the small size of structures inspected, which may affect measurement precision and accuracy, though we validated our measurements through tests of intra-rater reliability. Fifth, due to the time-intensive nature of data collection, case evaluation was performed by one neuroradiologist.

In conclusion, the present study demonstrates non-invasive visualization of the dorsal and ventral lymphatic systems of the brain in living humans using intrinsic signals produced by mLVs without the need for any external tracers. Moreover, we visualized direct communications of CN lymphatic flow with cervical LNs. Provided that our findings are replicated in additional healthy subjects, this noninvasive technique could be used to evaluate the MVLs system and function of the brain, which may allow for new approaches in diagnosis or treatment of neurological disorders such as traumatic brain injury, Alzheimer disease, multiple sclerosis. Like the peripheral lymphatic system, the drainage systems of the CNS appear highly complex. A complete understanding of the full dynamics of CNS lymphatic efflux will rely on a synthesis of findings from multiple techniques, as each technique available to inspect these systems has advantages and disadvantages.

## Methods

**Ethical permissions**. This retrospective study was approved by the local institutional review board (IRB201902528) and conducted with data from the Picture Archiving and Communication Systems (PACS/ Visage 7.1.14, Visage Imaging®). Since no new patient information was collected for the study, and all data in PACS are de-identified, the IRB approved the study as a retrospective review with a waiver for patient consent.

**Experimental design, patient selection, inclusion and exclusion criteria**. The 3D fluid-attenuated inversion recovery (FLAIR) series has been recently employed in seizure imaging protocols for evaluation of subtle parenchymal lesions. The cohort of patients in this study included those scanned with a 3D T2-FLAIR technique in a new generation 3 T MR scanner (Prisma, Siemens, Erlangen, Germany) using a 32-channel array head coil within a 13 month range due to clinical history of epilepsy or suspicion of epilepsy. Patients were identified from our PACS database (Visage 7.1.14, Visage Imaging®). Patients meeting all of the following criteria were included: (1) MR imaging with 3D FLAIR series without contrast injection, (2) the availability of at least one MR imaging study of diagnostic quality from the 3D FLAIR series, and (3) MR imaging done with a 3 T Prisma scanner (Siemens, Erlangen) with specific MR parameters. Patients with age-related changes such as minimal atrophy or mild white matter disease were not excluded. Exclusion criteria included (1) IV gadolinium injection before 3D FLAIR series, (2) significant motion artifacts on 3D T2-FLAIR scan, rendering quality insufficient to recognize the dural and subarachnoid spaces, and (3) previous intracranial surgery, significant atrophy, mass lesion, stroke, status epilepticus, cytotoxic edema on diffusion weighted imaging, focal parenchymal lesion, significant small vessel disease, or congenital developmental abnormalities. Intracranial surgery, parenchymal lesions, encephalomalacia, and stroke can destroy the parenchyma and might decrease visualization of normal lymphatic structures, and therefore these patients were excluded. Status epilepticus or seizure patients with cytotoxic edema are excluded as these conditions may increase or decrease glymphatic flow. Sedation may increase[40] or decrease[61] glymphatic flow and lymphatic flow as shown in animal studies. A total of 81 human subjects (45 females and 36 males) with a mean age of 41.7 (SD 20.4, range 15–80) years met the inclusion criteria and were included in this retrospective study. Details of inclusion and exclusion criteria and number of the patients from our cohort, including sex and age information, are provided in Fig. 2.

**MRI protocol**

*Phantom imaging and phantom MR imaging with a healthy human subject*. All phantom measurements were performed on a 3-T MRI scanner (Prisma, Siemens, Erlangen, Germany) using a 32-channel array head coil. We designed three sets of phantom studies using disposable plastic 10-mL syringes filled with diluted bovine serum albumin (BSA) protein (Sigma–Aldrich, St. Louis, MI). The stock BSA (500 mM solution) was diluted with double-distilled water (ddH2O) in two series of dilutions (5, 10, 20, 40, 80, 160, 320, 640, 1280 and 1200 mg/dl; Fig. 1a) and (7.5, 15, 30, 60, 120, 240, 500, 1000, 2000 and 4000 mg/dl; Fig. 1c, e) to simulate the protein concentration of lymphatic fluid (Fig. 1). The measured syringes were surrounded by agar within a plastic container for the agar to minimize motion by vibration and eliminate the influence of surrounding air-induced artifacts. The phantom was warmed up to 30 °C in a warm water bath before scanning.

First, we used a round phantom from which images were obtained six times in 3D-T2 FLAIR by varying the TE at 273, 300, 386, 499, 550 and 601 ms. Details can be seen in Fig. 1a. The other parameters were set as follows, field-of-view 230 × 230, matrix 512 × 512, 0.9 mm sections, TR/TI = 5000/1800 ms, nonselective inversion pulse, echo-train length 246, bandwidth 750 Hz/pixel, acceleration factor 1, acquisition time 4 min, 44 s, with fat saturation. In the phantom images, we measured the average SI within the center of each albumin solution syringe. We normalized the mean SI relative to the agar, defining normalized SI values as the SI value of each albumin solution divided by the SI value of agar at each corresponding TE value. Normalized SI values were compared for each albumin concentration at each TE value (Fig. 1b).

Second, two sets of 3D-T2 FLAIR images were obtained from a rectangular phantom at TI values of 1400 and 1600 ms (Fig. 1c). Other sequence parameters were as follows: field-of-view 230 × 230, matrix 512 × 512, 0.9 mm sections, TR/TI = 5000/387 ms, echo-train length 246, bandwidth 750 Hz/pixel, acceleration factor 1, acquisition time 4 min, 33 s, with fat saturation. In these phantom images, we measured the average SI within the center of each albumin solution syringe. We normalized the mean SI relative to the agar, defining normalized SI values as the SI value of each albumin solution divided by the SI value of agar at each corresponding TI value. Normalized SI values were compared for each albumin concentration at each TI value (Fig. 1d).

Last, the small phantom and a healthy adult male human subject were scanned simultaneously. The phantom was inserted above the head of the subject (Fig. 1g). While maintaining phantom temperature at between 22 and 30 °C, MR mages were obtained from whole-brain T2-FLAIR scans with fat saturation (Sagittal 3D acquisition, SPACE sequence, field-of-view 230 × 230, matrix 512 × 512, 0.9 mm sections, TR/TE/TI = 5000/387/1400 ms, nonselective inversion pulse, echo-train length 246, bandwidth 750 Hz/pixel, acceleration factor 1, acquisition time 6 min, 14 s, with fat saturation). In the phantom image, we measured the average SI within the center of each albumin solution syringe. Additionally we measured the parasagittal lymphatic signal along the sagittal sinus from the healthy human brain images along with the phantom.

**Patients' MR imaging**. All clinical MR imaging was performed on a 3 T Prisma scanner (Siemens, Erlangen) with a multichannel head coil (32-channel head coil). For each patient, images were obtained from whole-brain isotropic T2-FLAIR scan with fat saturation (Sagittal 3D acquisition, SPACE sequence, field-of-view 230 × 230, matrix 512 × 512, 0.9 mm sections, TR/TE/TI = 5000/387/1800 ms, nonselective inversion pulse, echo-train length 246, bandwidth 750 Hz/pixel, acceleration factor 1, acquisition time 5 min, 42 s, with fat saturation). All patients were scanned in the supine position.

### Image analysis

*Clinical MR imaging*. Analysis of all images was performed by M.S.A., a neuroradiologist with 20-years of experience. Sagittal isotropic 3D T2-fluid FLAIR images were reformatted into coronal, sagittal and axial 0.9-mm slices in the Picture Archiving and Communication Systems (PACS/ Visage 7.1.14, Visage Imaging®). The neuroradiologist was blinded to patient's information including age and sex.

During our clinical studies, we noticed that hyperintense signals within the parasagittal and perisinus dural structures in 3D T2-fluid FLAIR images without Gd injection corresponded with sites of MVLs structures described in previous animal and human studies. We also noticed similar patterns of hyperintensities in the ventral part of the brain, particularly within the dura at the level of the cranial nerve orifices.

First, we noted the presence or absence of hyperintense signals at these locations including the dorsal and ventral aspects of the brain. Later, maximum thickness and maximum SI was measured along the anterior, middle and posterior sagittal sinus, sinus rectus, confluence, bilateral transverse sinus, bilateral transverse sinus-sigmoid sinus junction, bilateral sigmoid sinuses, bilateral jugular veins, and posterior aspect of the foramen magnum. These structures are accepted as the dorsal lymphatic system. Single point ROIs representing MLVs were expertly and manually drawn by the neuroradiologist. ROIs represented areas of maximum SI in the dorsal MVLs regions derived from multiplanar reconstruction of the isotropic 3D FLAIR images. Later, maximal thickness of MLVs measurements were derived using the same approach. A depiction of our measurement technique for dorsal and ventral structures are provided (Supplementary Fig. 4). Each anatomical structure was measured in the projections that best represented the max SI and perpendicular max thickness. For example superior sagittal sinuses (SSS) are generally measured in the coronal views, but sometimes on sagittal series the confluence in sagittal or coronal views, and the dura at the level anterior cranial fossa in coronal views. We did not necessarily always acquire paired SI and thickness values for a given ROI in a given patient from the same individual image slice from that patient.

In animal studies, ventral lymphatic pathways are much more pronounced relative to the dorsal pathways, but in humans, this ventral lymphatic flow can be reliably imaged only along the dura and olfactory nerve in the anterior cranial fossa. Previous human studies using heavily T2-weighted fluid-attenuated inversion recovery (hT2w-FLAIR) MRI magnetic resonance imaging revealed that inner ear structures, optic nerves, and Meckel caves show significant gadolinium absorption, based on measurements obtained before, and 3 h and 24 h after intravenous administration of gadolinium-based contrast agent (GBCA)[41]. We noticed a similar pattern of signal enhancement in the ventral dura around the major skull base foramina without gadolinium injection in our 3D-T2-FLAIR sequence parameters. Specifically, we managed to visualize increased dural signal intensity around the major cranial nerves, anterior cranial fossa dura, olfactory groove adjacent to the optic nerves, around the Meckel cave orifices, in the Meckel cave, around the internal auditory canal, and within the jugular foramen. These signals might represent the major components of the ventral lymphatic system in the human brain. Again, dural blood vessels and CSF were both dark in our 3D-T2-FLAIR sequences, whereas mLVs were white linear areas surrounding the dura at the level of the major cranial foramina. Maximum thickness and maximum SI were also measured in the dural/paradural structures along the anterior cranial fossa, optic groove, diaphragma sellae, bilateral dura at the orifice of the Meckel caves, bilateral dura in the Meckel caves, bilateral dura at the orifice of the internal auditory canals, and bilateral dura at the level of the orifices of jugular foramen. These structures are accepted as the ventral lymphatic system. A few representative approaches to measurement are shown below. Each anatomical structure is measured in the projections that depicts the maximum signal intensity and perpendicular to the maximum thickness as detailed below. We did not necessarily always acquire paired SI and thickness values for a given ROI in a given patient from the same individual image slice from that patient.

In the same manner described for other structures, maximum SI was recorded in the bilateral extracranial IX–XII cranial nerve complexes at the level of the skull base superior to the cervical lymph nodes (LNs), and bilateral extracranial IX–XI cranial nerve complex at the mid-neck level inferior to the cervical LNs. Max SI and short diameter of the bilateral retropharyngeal LNs and bilateral dominant vascular LNs were recorded.

The absence or presence of connections between ICAs-jugular veins-CN IX–XII complex in the vascular space, and cervical LNs, including retropharyngeal and vascular LNs, was recorded. As reference points for internal normalization for age and sex comparisons, max SI values of bilateral superficial temporalis muscle at the level of frontal skull base, bilateral centrum semiovale were measured using single voxel-sized ROI.

Overall mean dorsal SI and thickness values were calculated using six variables which were available in all of the patients: anterior sagittal sinus, middle sagittal sinus, sinus rectus, confluence, and bilateral sigmoid sinus-transverse sinus junctions. Overall mean ventral SI and thickness values were calculated using ten regions: anterior cranial fossa dura, optic groove dura, bilateral Meckel's orifice dura, bilateral dura in the Meckel's caves, bilateral dura at the orifice of the internal auditory canals, and bilateral dura at the level of the orifices of jugular foramen (Supplementary Fig. 4).

The aging process induces changes in structure and function of systemic lymphatic networks[60,62–64]. Muscle cell atrophy, elastic element destruction, and aneurysm-like formations have been identified within aged systemic lymphatic vessels[60,62–64]. Animal studies have documented a severe decrease in pump action of these vessels with age, mainly due to decreased contractile speed[62,63], and likely secondary to age-related enlargement of the lymphatic vessels[63]. We expect similar changes in the intracranial lymphatic systems with aging. Sex related changes were also compared in this study. ROI SI values were corrected based on superficial temporalis muscle SI measurements in the comparison by age because mean SI values of this region were similar in both age groups (Supplementary Table 8). For comparisons by sex, ROI SI values were corrected based on central white matter SI values based on similarities in these values between sexes (Supplementary Table 9). The dorsal/ventral SI and dorsal/ventral thickness ratios were calculated in order to assess the balance between these systems and the changes related to age and sex. Also, the SI/thickness ratio was calculated within structures of the dorsal and ventral compartments for the comparison by age and sex. We postulate that SI/thickness ratios of the dorsal and ventral compartments represent the efficacy of the lymphatic drainage in these systems (i.e., strength of pump activity and contractile function). Comparisons were made by age and sex between percentage of visible regions in the posterior dorsal areas, mean ventral and dorsal SI and thickness values; and retropharyngeal and vascular LN SI and short axis values.

*Volumetric analysis for intracranial cavity, tissue volumes and CSF volumes*. Our Epilepsy MR protocol includes 3D-T1-weighted gradient-echo (MPRAGE-magnetization-prepared rapid acquisition of gradient echo, acquisition matrix 256 × 256, isotropic resolution 0.9 mm, 192 slices, repetition time [TR]/echo time [TE]/inversion time [TI] = 1720/2.11/865 ms, flip angle 9, acquisition time 5 min 45 s). This volumetric series was available for each patient. A new software pipeline for volumetric brain analysis, provides automatically volumetric brain information at different scales in a very simple web-based interface (volBrain version 1.0 release 04/03/2015) (https://www.volbrain.upv.es)[65]. The volBrain pipeline is a set of image-processing modules that modifies input images into a specific geometric and intensity space allowing for segmentation of different structures/tissues of interest. This pipeline includes the following steps: (1) spatially adaptive nonlocal means denoising, (2) rough inhomogeneity correction, (3) affine registration to Montreal Neurological Institute (MNI) space, (4) fine statistical parametric mapping–based inhomogeneity correction, (5) intensity normalization, (6) nonlocal intracranial cavity extraction (NICE), (7) tissue classification, (8) nonlocal hemisphere segmentation, and (9) nonlocal subcortical structure segmentation. The volBrain system provides volumes/segmentations and structure asymmetry ratios for different measures the intracranial cavity (sum of all white matter, gray matter, and cerebrospinal fluid), and tissue volumes (including white matter, gray matter, and cerebrospinal fluid volumes)[65]. We obtained the total intracranial volume, gray matter, white matter and CSF volumes, as $cm^3$ and percentage of each.

**Statistical analysis**. Statistical analysis was performed with SPSS 20.0 (IBM, Armonk, New York). Categorical variables were expressed as frequencies and percentages and continuous variables as mean (SD) or median (IQR). Categorical variables were compared using chi-square tests and continuous variables were compared using paired samples $t$, Wilcoxon, Student $t$, or Mann–Whitney $U$ tests. Correlations were assessed using Pearson or Spearman correlation tests and also adjusted for age, sex, and intracranial volume using partial correlation tests. In the first ten cases, measurements were performed on two different times for each thickness and SI value in order to assess intra-rater reliability by intraclass correlation coefficient analysis using two-way mixed-effects model and absolute agreement[66]. Repeated measures were made blind to each other. ROC curve and binary logistic regression analyses were used to define the optimal age cut-off for discriminating between lower than mean dorsal and ventral SI/thickness ratios. Statistical significance was established at $p < 0.05$.

**Reporting summary**. Further information on research design is available in the Nature Research Reporting Summary linked to this article.

## Data availability

The minimum dataset necessary to interpret, verify and extend the research in this article is accessible within the manuscript and its Supplementary Information. Source data are provided with this paper. Anonymized images and any additional raw data in this work are available from the corresponding author upon reasonable request. Source data are provided with this paper.

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

## Acknowledgements

O.A. is an Alzheimer's Association Research Fellow and supported by MUSC's Specialized Center of Research Excellence (SCORE) Career Enhancement Core (CEC) Scholarship. We thank the MRI technologists at UF Health Shands Hospital for their invaluable assistance that they generously provided throughout this project. We also thank professor Michael D. Weiss, M.D., of the University of Florida College of Medicine Department of Pediatrics for his assistance in reviewing the manuscript.

## Author contributions

M.S.A. designed the study. M.S.A., G.S. and O.A. coordinated and directed the project. M.S.A. and I.S.T. collected clinical and imaging data. M.S.A., F.T., G.S., M.B., and O.A. wrote the report. M.S.A., F.T., G.S., M.Z., and O.A. provided scientific direction. M.S.A., G.S., and I.S.T. performed image analysis. M.S.A., F.T., and O.A. performed data analysis. F.T. performed statistical analysis.

## Competing interests

The authors declare no competing interests.
