## [Peer Review File · Nature Communications]

Reviewers' comments:

Reviewer #1 (Remarks to the Author):

This study uses FLAIR to non-invasively to systematically image the dorsal and ventral lymphatic systems of the brain in patients. The study also proves evidence for a fluid communication pathway between ICA and CNs lymphatic flow with cervical LNs. Veery nice schematics are included to explain the findings.

Critique:

1) The abstract is too vague. The abstract should not review the absence of data in the field but state what was done and what the analysis shows. The abbreviation "FLAIR" should be included so it is clear what technique was utilized. The number of subjects should be listed. A statement like "Our findings also represent age and sex related differences in these putative mLVs" should be changed to a clear statement on what was actually observed. In other words, thee abstract should summarize how this study brings the field forward.

2) Flair imaging does not have a cellular resolution. Thus, it is not possible to distinguish between the smooth muscle cell layer and lymphatic vessels surrounding the ICA. The authors seems to forget this when referring to the iPAD concept. This model is based on histological analysis. More recent studies have shown that first tracers enter the smooth muscle layer during cardiac arrest (Mestre et al., Nat comm 2018, TINS, 2020). In vivo the tracers are contained within fluid-filled perivascular spaces and redistribute to the smooth muscle cell layer upon cardiac arrest. The lack of in vivo data supporting the iPAD concept make it in particular troublesome that this manuscript again and again emphasizes the strength of in vivo imaging and then uncritically refers to the iPad concept. Especially because many groups have raised critique. The extracellular space of smooth muscle tissue is the relative smallest of all tissue, constituting only 10% of the total tissue volume which will impose a high resistance toward fluid flow.

3) Age dependent changes in mouse lymphatics have been described. The authors should discuss this topic in more details and point the conclusion out in the abstract.

Reviewer #2 (Remarks to the Author):

Albayram et al uses 3T MR to investigate the lymphatic network in human CNS and cervical region. The paper characterizes ventral lymphatics in the CNS and surrounding cranial nerves. Lymphatic structures associated with the CNS is a relatively new topic which is of interest to the neuroscience community. However, the study is limited by only using one method for identification of lymphatic structures without much verification and the lack of functional studies or link to disease.

Major points

The authors state that this is the first to describe how FLAIR MR can be used for detection of lymphatic structures. It is an issue that the sequence is used both presented as novel method without extensive validation using established methods, however, is simultaneously used to identify new or under-appreciated cervical lymphatic structures.

The description of the method for lymphatic tissue identification is not easy to follow. Other structures, e.g. the corpus callosum also shows hyperintense signal on the FLAIR images. Was there a cut-off value or how was the FLAIR sequence used to identify lymphatic structures? The study appears to combine what are already known lymphatic structures with supposed ones (or less known ones) without using clear-cut parameters for defining lymphatic tissue.

The lymphatic structures for example surrounding the olfactory nerve are not novel more references are needed regarding what is already known of CNS-associated lymphatics.

The Eide group has demonstrated that intrathecally injected contrast is drained to cervical lymph nodes and also makes a plausible argument for dorsal lymphatics. (Ringstad & Eide Nature Communications 2020 doi: 10.1038/s41467-019-14195-x among other papers)

Kipnis & Reich used IV contrasts of different sizes to visualize the dorsal intracranial lymphatics in monkeys and humans (2017). The Nedergaard lab at University of Rochester showed intrathecal tracer distributing to the eye in mice (2020), Johnston at University of Toronto observed intrathecal tracer surrounding the olfactory nerve and at the lymphatic endothelium structures in the nasal area (2003). This year, basal lymphatic structures in mice were also published by Ahn et al (Nature). The manuscript would benefit from trying to expand the work and novelty factor, perhaps with comparing a control to a patient group, such as MS, where extracranial lymphatics are expected to be increased based on findings from animal models.

The study is also generally quite under-referenced, example: 'Arachnoid granulations and villi of the intracranial and spinal venous sinuses are specialized to filter smaller, acellular particles. Most CSF drains directly into the bloodstream through these structures.' The evidence for this is debatable and needs references.

In summary, although the topic is of great interest, the method appears to need more validation in order to make the conclusions presented. The novelty factor could be improved and more references are needed.

Responses to the Reviewers' Comments:

We deeply appreciate the careful analysis and constructive suggestions on improving our manuscript by the reviewers. **Reviewer 1** indicated that “*This study uses FLAIR to non-invasively systematically image the dorsal and ventral lymphatic systems of the brain*” and “*also provides evidence for a fluid communication pathway between ICA and CNs lymphatic flow with cervical LNs*”. It has also been specified that “*very nice schematics are included to explain findings*” but raised questions about the structure of the manuscript and the methodology. **Reviewer 2** pointed out the quality of our study and importance of our report to the neuroscience community but raised concerns about the methodological limitations and its putative functional readouts.

Reviewer #1, Comment 1:

The abstract is too vague. The abstract should not review the absence of data in the field but state what was done and what the analysis shows. The abbreviation “FLAIR” should be included so it is clear what technique was utilized. The number of subjects should be listed. A statement like “Our findings also represent age and sex related differences in these putative mLVs” should be changed to a clear statement on what was actually observed. In other words, the abstract should summarize how this study brings the field forward.

Response to Reviewer #1, Comment 1:

We fully agree with the reviewer that our initial abstract did not address the major aspects of the article. We have now rewritten the abstract based on the suggestions of the reviewer.

Reviewer #1, Comment 2:

Flair imaging does not have a cellular resolution. Thus, it is not possible to distinguish between the smooth muscle cell layer and lymphatic vessels surrounding the ICA. The authors seem to forget this when referring to the iPad concept. This model is based on histological analysis. More recent studies have shown that first tracers enter the smooth muscle layer during cardiac arrest (Mestre et al., *Nat comm* 2018, *TINS*, 2020). *In-vivo* the tracers are contained within fluid-filled perivascular spaces and redistribute to the smooth muscle cell layer upon cardiac arrest. The lack of *in-vivo* data supporting the iPad concept make it in particular troublesome that this manuscript again and again emphasizes the strength of in vivo imaging and then uncritically refers to the iPad concept. Especially because many groups have raised critique. The extracellular space of smooth muscle tissue is the relative smallest of all tissue, constituting only 10% of the total tissue volume which will impose a high resistance toward fluid flow.

Response to Reviewer #1, Comment 2:

We are grateful to the reviewer for bringing up this important point and apologize for our oversight. Certainly, FLAIR images cannot show smooth muscle cells or lymphatic vessels in the wall of the artery. In our study, we detected prominent signal around the petrous internal carotid artery (ICA) at the level of the skull base but not in intracranial arterial structures including cavernous ICAs. Indeed, this signal became mostly imperceptible inferior to the cervical lymph nodes, a pattern consistent with interstitial fluid (ISF)-lymphatic flow around the artery. We saw no prominent signal at the level of the cavernous ICAs superiorly, and our technique cannot detect direct continuous signal from intracranial arteries to petrous ICA. As pointed out by the reviewer, we have now added an additional figure (Fig. 6C) in our revised manuscript, which displays no signal around the cavernous ICA and appears at the level of petrous ICAs. We merely observe the petrous ICA signals, upper cervical ICA signals, and lymph node connections. Based on our main observations, lymphatic fluid flows into walls of dural venous sinuses, the meningeal lymphatics, and along cranial nerves, which fits closely with the Glymphatic Theory instead of the iPad concept. According to the iPad concept, ISF mainly drains towards intracranial arteries which then drain into the cervical lymph nodes, not via meningeal lymphatics which drain only CSF (Engelhardt B. et al., *Acta Neuropathol* 2016, 132:317-338). In the Glymphatic Theory, the most accepted theory at this point, solute- and macromolecule-rich ISF-CSF drains via meningeal lymphatics and eventually reaches cervical lymph nodes (Mestre H. et al., *Trends in Neurosciences* 2020, 43(7):458-466). We enclose two illustrations below retrieved from each of these prior publications, respectively.

Our observations also support the hypothesis that protein and solute rich ISF drain mainly along the dorsal and ventral lymphatic systems but not exclusively along the main intracranial large arteries and connected cervical ICAs. Moreover, findings in our study are consistent with those of previous studies. Aspelund A., et al. was among the first to describe dural lymphatics that drain brain interstitial fluid and macromolecules within animal models (Aspelund A. et al., *J Exp Med.* 2015, 212(7): 991-999). Remarkably, they detected lymphatic vessels adjacent to the ICA immediately below the skull. Additionally, they identified that brain ISF drains into the lymph node via the efferent carotid lymphatic vessels in the neck (as shown in adapted figure).

Our observations also support the hypothesis that protein and solute rich ISF drain mainly along the dorsal and ventral lymphatic systems but not exclusively along the main intracranial large arteries and connected cervical ICAs. Moreover, findings in our study are consistent with those of previous studies. Aspelund A., et al. was among the first to describe dural lymphatics that drain brain interstitial fluid and macromolecules within animal models (Aspelund A. et al., *J Exp Med.* 2015, 212(7): 991-999). Remarkably, they detected lymphatic vessels adjacent to the ICA immediately below the skull. Additionally, they identified that brain ISF drains into the lymph node via the efferent carotid lymphatic vessels in the neck (as shown in adapted figure).

Significant Signal along Cavernous ICAs and Inversely Significant Flow along Petrous Internal Carotid Artery (Green arrows) and no flow in the cavernous ICA (White arrows). (adapted from the Figure 4).

Clapham R., et al. showed the direct relationship between the cervical ICAs and cervical lymph nodes in a human autopsy study. We believe our periarterial signal around the ICAs represents ISF flow from brain and lymphatic vessels that connect with lymph nodes, which is distinctly different from the iPad concept. Instead, our findings suggest that ISF/lymphatic signal around petrous ICAs at the level of the skull base may be derived primarily from the skull base structures such as the sympathetic plexus around the ICA. We could not see continuous flow from intracranial ICA to petrous ICA, which may be due to technical limitations, though we suspect this is less likely. Further studies would be needed to test this hypothesis. Overall, our findings suggest that drainage of ISF occurs along multiple structures including cranial nerves, dural venous sinuses and the petrous ICA, with the ICAs conveying only a fraction of the total drained fluid volume, possibly <15%. This concept is conveyed in the newly added Fig 9 of our manuscript. We would be happy to provide other necessary clarifications if needed.

We rewrote our discussion regarding ICA signals to provide further clarification. Additionally, we support our claims with the newly added Fig 4c and schematically depict our model in the newly added Fig 9.

Reviewer #1, Comment 3:

Age dependent changes in mouse lymphatics have been described. The authors should discuss this topic in more details and point the conclusion out in the abstract.

Response to Reviewer #1, Comment 3:

We fully agree with our reviewer that a more elaborate discussion of the age-dependent changes in mouse brain lymphatics would strengthen the manuscript. We have now rewritten the aging section of the manuscript and carefully discussed current animal and human studies related to age-dependent changes in brain lymphatics. Additionally, we included the aging results in the abstract.

Reviewer #2, Comment 1:

The authors state that this is the first to describe how FLAIR MR can be used for detection of lymphatic structures. It is an issue that the sequence is used both presented as novel method without extensive validation using established methods, however, is simultaneously used to identify new or under-appreciated cervical lymphatic structures.

Response to Reviewer #2, Comment 1:

We fully acknowledge the reviewer's desire for more scientific proof and validation. 3D-FLAIR MR imaging is a novel technology and has only recently been used for imaging of the brain lymphatics with intravenous or intrathecal injection of gadolinium-based contrast agents. Absinta M et al. was one of the first groups to use 2D and 3D T2 FLAIR MR imaging and T1-weighted black blood techniques in pre- and post-contrast agent administration (Absinta M. et al., *Elife* 2017, 6:e29738). Particularly, Figure 1 in Absinta M et al. demonstrates well that 2D FLAIR MR dural lymphatic signals in the human brain can be seen before contrast administration but becomes more prominent following contrast administration. Moreover, Ringstad G et al. and Zhou Y et al. also used 3D T2-FLAIR and T1-weighted black blood imaging techniques for detection of the meningeal lymphatics (Ringstad G. et al., *Nat Commun.* 2020, 11:354; Zhou Y. et al., *Ann Neurol.* 2020, 87:357-369). In particular, coronal 3D T2-FLAIR images before contrast agent administration in Figure 1 of Ringstad G et al. reveal a signal intensity identical to those of our images at the level of the mid-sagittal sinus (as shown in adapted figure below). We enclosed related illustrations below retrieved from each of these prior publications, respectively.

Coronal 3D T2-FLAIR MR image visualization at prior contrast agent administration. (adapted from Absinta M. et al., *Elife* 2017, 6:e29738).

Sagittal and coronal T2-FLAIR MR images, respectively, show the typical longitudinal and lateral extension of parasagittal dura (PSD) with high signal (arrows) at prior contrast agent administration (adapted from Ringstad G. et al., *Nat Commun.* 2020, 11:354).

Sagittal 3D T1 (A1, 2) and 3D T2-FLAIR MR (B1, 2) image visualization at prior contrast agent administration. (adapted from Zhou Y. et al., *Ann Neurol.* 2020, 87:357-369).

Our non-contrast 3D T2-FLAIR MR imaging parameters permitting detailed visualization of dorsal putative meningeal lymphatic vessels (adapted from the Figure 2).

In this study, we observed significant signal intensity on 3D isotropic FLAIR series without contrast administration in the locations of lymphatic structures described in previous human MR studies along the sagittal sinus and anterior cranial fossa in all 81 human subjects studied. As we did not use contrast, 3D FLAIR imaging is able to see signals in the locations of lymphatic tissues due to high protein and macromolecular concentration characteristic of lymphatic fluid.

To better validate this idea, we have now performed a series of MRI phantom studies which provide a more concrete explanation of these putative brain lymphatic signals without the need for contrast administration. Ten albumin solutions of varying concentrations (5, 10, 20, 40, 80, 160, 320, 640, 1280 and 1200 mg/dl; Fig. 1a) and (7.5, 15, 30, 60, 120, 240, 500, 1000, 2000 and 4000 mg/dl; Fig. 1c, e) were imaged to simulate the protein concentration of lymphatic fluid using 3D T2-FLAIR (Fig. 1).

First, we used a round phantom from which images were obtained six times in 3D-T2 FLAIR by varying the TE at 273, 300, 386, 499, 550 and 601 msec. In this study, increasing TE values resulted in elevated the relative SI across the protein samples analyzed, with the highest protein-agar-ratio observed at TE of 601 msec (Fig a, b). Mild signal increase relative to ddH₂O can be seen from 10-20 mg/dl, but a decrease in signal back toward that of ddH₂O is observed at concentrations of 40-160 mg/dl levels. Further increase in protein concentration from 320 to 1280 mg/dl results in a marked increase in signal (Fig 1b). Furthermore, increasing TI values also resulted in increasing relative SI across the protein samples analyzed, with the higher protein-agar-ratio observed at TI of 1600 msec (Fig. 1c, d). Additionally, signal decrease back toward that of ddH₂O occurred at concentrations of 60 mg/dl levels at TI of 1400 msec and at 240 mg/dl levels at TI of 1600 msec. Drastic signal changes were therefore observed across the protein solutions at different MR parameters in our phantom studies. We assume proper MR sequence parameter selection is critical to specific visualization imaging of brain lymphatics.

We further found that signal within expected regions of dural lymphatics corresponded to protein concentration ranging from 2000-4000 mg/dl with 3D-T2-FLAIR sequence parameters set at TR/TE/TI = 5000/386/1400 msec, suggesting that regions of high protein and macromolecular concentration are present in putative lymphatic structures along the sagittal sinuses (Fig. 1e-h). The signal-to-noise-ratio within the region of dural lymphatics of the healthy subject indicates that the protein concentrations of putative dural lymphatics are significantly higher

than those of cerebrospinal fluid (CSF) and may allow for detection of the meningeal lymphatic pathways without the need the contrast administration. Notably, we could not detect significant signals within regions of CSF, where protein concentration (15 to 60 mg/dL in adults) is lower than that of lymphatic fluid (Fig. 1i, j). Additionally, we provided mean SI measurements of all major structures including the cervical lymph nodes (Fig. 1j), demonstrating that this novel T2-FLAIR sequence's parameters are sensitive to lymphatic fluid and lymphatic tissue without the need for contrast agents. We now included a completely new study and figure (Figure 1) on this revised manuscript, which provides protein concentration versus TE values, mean signal intensity of the head and neck structures, and protein solution versus signal changes at the selected TI value, and also provides a comparison of standard protein solution and live human putative lymphatic fluid. It appears that putative lymphatic fluid has a very high protein concentration. We believe this new phantom study and added figures provides better validation of our technique. New sections are provided in the methods and results of the manuscript describing the phantom studies.

In addition, we fully acknowledge the reviewer's concerns that small lymphatic connections between the cranial nerves, ICA and lymph nodes have not been previously shown in live humans non-invasively with MR imaging. Our study is the first to provide non-invasive visualization of putative brain lymphatics along the venous

structures, cranial nerves, and cervical ICAs extending to lymph nodes in living humans, with direct connections to cervical lymph nodes. We improved the caption and coloration of Fig 4 to facilitate understanding of our claims. We also included three additional examples in supplementary figures to demonstrate connections between cranial nerves and vascular structures to lymph nodes in living humans. A schematic representation is now provided in Figure 9 of our overall model.

We hope the MRI phantom studies performed for validation of our technique addresses the reviewer’s concerns and have included pertinent modifications in the discussion section and figures.

Reviewer #2, Comment 2:

The description of the method for lymphatic tissue identification is not easy to follow. Other structures, e.g. the corpus callosum also shows hyperintense signal on the FLAIR images. Was there a cut-off value or how was the FLAIR sequence used to identify lymphatic structures? The study appears to combine what are already known lymphatic structures with supposed ones (or less known ones) without using clear-cut parameters for defining lymphatic tissue.

Response to Reviewer #2, Comment 2:

We fully agree with the reviewer that the description of the method for lymphatic signals identification was not clear. In line with the suggestion, we have reframed the entire method and result sections.

“The Other structures, e.g. the corpus callosum also shows hyperintense signal on the FLAIR images”.

We thank the reviewer for this important comment. In Fig. 2a, the linear signal under the corpus callosum is the ependymal signal, not related to corpus callosum. The corpus callosum is the one of the largest white matter structures in the brain and significantly darker in the T2-FLAIR MR images. Average SI of white matter can be

seen in Fig. 1i, j. Additionally, Fig. 3a, c verifies the low signal of the corpus callosum. The etiology of the high periventricular ependymal signal is unknown but can be seen with ageing and normal pressure hydrocephalus. We did not analyze the ependymal signal in our study.

“Was there a cut-off value or how was the FLAIR sequence used to identify lymphatic structures?”

We thank the reviewer for this important comment. No, we did not use any cut off value. We try to describe the anatomical structures and find their average signal intensity and thickness. Usage of the cut off values is difficult here because wide variation is present between anatomical structures. This is especially true in the dorsal system. The dependent portions of dorsal lymphatic structures such as the posterior sagittal sinus, transverse sinus, and sigmoid sinuses show thin lymphatic signal and low signal intensity values. These changes are most likely related to gravity-induced compression of these structures by the occipital lobe due to the body’s supine position. Instead, we believe that the signal intensity and thickness values would be changed with upright MR imaging if possible. Our search strategy for lymphatic signals was dependent on data from previously published animal studies. It has been shown in animal studies that all sinus walls in the intracranial region have lymphatic signals in the wall-dura and the lymphatic signal always follows the cranial nerves and skull base within the ventral portion of the brain. Naturally the human brain requires gateways for flow of ISF/lymphatic fluid between intracranial spaces and the neck region. Indeed, neural foramina are among the expected exit points in the otherwise closed human skull base. Therefore, we chose dural signals at the orifices of major neural foramina which connect intracranial spaces and neck spaces. With reference to our response to the previous comment, we believe T2-FLAIR MR images cannot visualize the lymphatic tissue itself, but rather can detect the lymphatic fluid within the vessel due to sensitivity of this technique to elevated protein concentration. We validate our technique and establish a logical explanation of these putative brain lymphatic signals through a series of MRI phantom studies (detailed in previous comment). Additionally, Mehlem et al. demonstrated that FLAIR MR imaging is very sensitive to fluids with increased protein content, such as those in the subarachnoid spaces during inflammatory responses, making it useful in detecting meningitis, subarachnoid hemorrhage, and other pathology affecting the dura and subarachnoid spaces (Melhem ER. Et al., *AJR Am J Roentgenol.* 1997, 169(3):859-62). In our study, we demonstrated that 3D FLAIR signals generated from different protein solutions directly related to MR sequence parameters, such as TE values. Relative SI from the protein solutions significantly increases with increasing TE

Subtle gadolinium leak can be seen 31 minutes after the intravenous injection of gadolinium-based contrast agent in the anterior cranial fossa (adapted from Absinta M. et al., *Elife* 2017, 6:e29738).

Putative lymphatic signal is more prominent most likely related internal structures, without minimal leak of the gadolinium-based media (adapted from the Figure 3).

(Fig. 1a, b from phantom study), therefore clear visualization of brain lymphatic requires optimization of MR parameters.

Human MR imaging studies were able to detect the subtle lymphatic signals only in parasagittal dura with intravenous or intrathecal injection of gadolinium-based contrast agents. Moreover, human autopsy studies were also limited to parasagittal dura and anterior cranial fossa and olfactory nerve. Our study provides the first description of a continuous signal corresponding to putative lymphatic fluid compartments along the dorsal venous sinuses and ventral stations along the cranial nerves and petrous ICA, with each of these networks forming direct connections to each other and ultimately to deep lymph nodes in the neck.

Since our FLAIR technique eliminates the need for gadolinium-based contrast media, there was no need to delay imaging, or collect images at multiple timepoints, hypothetically allowing for full mapping of lymphatic fluid compartments in the human brain, including compartments not readily accessible to contrast media. Additionally, gadolinium-based studies only show the behavior of the metal, but not internal lymphatic signals. For example, a subtle gadolinium leak can be seen in the Absinta M et al. study in the anterior cranial fossa (as shown in the adapted figure below). In our study, we observed corresponding putative lymphatic signals in this region as well but with more prominent signal, likely due to the advantage of the 3D FLAIR technique and internally generated fluid signatures.

Reviewer #2, Comment 3:

The lymphatic structures for example surrounding the olfactory nerve are not novel more references are needed regarding what is already known of CNS-associated lymphatics.

Response to Reviewer #2, Comment 3:

We fully acknowledge the reviewer's desire for more references and background information regarding what is already known of CNS-associated lymphatics. We divided this answer into two parts, animal and human studies.

Animal Studies: Currently, CNS-associated lymphatics have been examined in detail in animal studies, especially mouse models. Louveau and Kipnis et al. showed CNS meningeal lymphatics along the sagittal sinus and the transverse sinuses via fluorescence immunolabeling with lymphatic endothelial cell targeted probes (anti-Lyve-1) (Louveau A. et al., *Nature* 2015, 523(7560):337-41). Furthermore, they used Evans blue dye and determined that meningeal lymphatic vessels communicate with the deep cervical lymph nodes (dCLNs), but not the superficial lymph nodes (as shown in the adapted figure below). Notably, no Evans blue was detected in the dCLNs 30 min after direct injection into the nasal mucosa, suggesting

that meningeal lymphatic vessels, but not nasal mucosa lymphatic vessels, represent the primary route for drainage of CSF-derived soluble and cellular constituents into the dCLNs during this time frame.

Next, Aspelund et al. showed CNS-associated lymphatics along the sagittal sinus, the transverse sinus, the sigmoid sinus, the retroglenoid vein, the rostral rhinal vein, and the major branches of the middle and anterior meningeal arteries (Aspelund A. et al., *J Exp Med.* 2015, 212(7): 991–999). They described that the ventral skull lymphatic vessels could be seen in the distal portion of several cranial nerves, including optic, trigeminal,

glossopharyngeal, vagus, and accessory nerves, exiting the skull along with the nerves (as shown in adapted figure).

They also detected lymphatic vessels in the dural lining of the cribriform plate, where some vessels passed through the skull into the nasal mucosa. Generally, lymphatic vessels were relatively scarce in the superior portions of the skull (dorsal region), whereas the base of the skull contained a more extensive lymphatic vessel network. Interestingly, only the lymphatic vessels at the base of the skull contained valves. Overall, they concluded that lymphatic vessels are present in the dura mater of the CNS and drain out of the skull via the foramina of the base of the skull alongside arteries, veins, and cranial nerves.

Finally, Ahn et al. detected meningeal lymphatic vessels (mLVs) along the superior sagittal sinus and transverse sinus, which had small diameters with a largely discontinuous lymphatic structure (Ahn JH. et al., *Nature* 2019, 572(7767):62-66). By contrast, basal mLVs running

along the petrosquamosal sinus (PSS) and sigmoid sinus had larger diameters and abundant protruding capillary branches with blunt ends consisting of typical oak leaf-shaped lymphatic endothelial cells (LECs) and lymphatic valves, which were similar to functional classic LVs. They concluded that basal/ventral mLVs are a hotspot for lymphatic drainage of CSF. Basal mLVs also showed direct connections with extracranial LVs through the skull foramina. As a result, these studies showed CNS-associated lymphatics along the dural venous sinuses, cranial nerves, and arteries in both dorsal and ventral regions in detail.

Human Studies: CNS-associated lymphatics in humans are described only along the superior sagittal sinus and in the anterior cranial fossa (Ringstad G. et al., *Nat Commun.* 2020, 11:354; Zhou Y. et al., *Ann Neurol.* 2020, 87:357-369; Wu CH. et al., *Ann Neurol.* 2020, online ahead of print). The autopsy studies only found the lymphatics along the sagittal sinuses (Absinta M. et al., *Elife* 2017, 6:e29738 and Goodman JR. et al., *Brain Behav Immun* 2018, 73:34-40). There are no significant data available regarding the ventral lymphatics in humans, except those of the olfactory nerve. Certain findings of human and animal studies, however, are in conflict. [Aspelund A. et al., *J Exp Med.* 2015, 212(7): 991–999; Ahn JH. et al., *Nature* 2019, 572(7767):62-66 Ma Q. et al., *Nat Commun.* 2017, 8:1434;] suggested that basal lymphatics are key for CNS-associated lymphatic drainage, but

Ringstad et al. identified strong tracer enrichment in parasagittal dura of the human brain suggesting that in humans the dorsal system is prominent whereas the ventral system plays only a minor role in lymphatic drainage. The findings of Ringstad et al. also contradict animal studies concluding that CSF outflow occurred predominantly

along cranial nerves through skull neuroforamina. These obvious discrepancies between observations in mice and human studies underline the importance of human translational research.

We believe that non-invasive imaging techniques to detect the internal signature of CNS-associated lymphatic fluid compartments and lymphatic flow are needed to explain the conflicts between human and rodent studies. Human and rodent brains show prominent structural differences that may explain these conflicts. The human brain features a large telencephalon associated with a prominent dural venous system. Rodent brains however demonstrate a larger ventral surface relative to their proportionately smaller telencephalon.

Lymphatics are small fluid filled structures that are difficult to evaluate in autopsy or pathology studies due to their fragility and propensity to collapse. Furthermore, microscope-based imaging techniques cannot penetrate the entire human brain and neck. MR imaging studies with intravenous or intrathecal injection of gadolinium-based contrast agents can cover the entire brain; however, gadolinium-based studies only show the behavior of the metal, not internal lymphatic signal. Dilution of gadolinium may prevent the visualization of the small lymphatic pathways. Also, gadolinium is toxic and always generates deposits in the body and brain, regardless of the injected amount. Multiple gadolinium injections are not safe for humans, making it difficult to find healthy controls for lymphatics studies, and are especially harmful to those with chronic neurological diseases, such as MS or Parkinson's. Therefore, we believe that our study is important because our technique shows full visualization of the dorsal and ventral hot spots with connections, including in the neck region. The ventral lymphatic system and its full extent in the brain have not yet been described in the live human. Direct connections from the dorsal and ventral systems to the cervical lymph nodes are also not described yet. Our technique will potentially improve future translational neuroscience studies on multiple sclerosis, Alzheimer's disease, and imaging of changes in CNS-associated lymphatics without any harm to the patients and healthy subjects, unlike techniques using gadolinium injection.

We fully agree with our reviewer that a more elaborate description of the animal and human studies of CNS-associated lymphatics would definitely strengthen the manuscript. We have added 20 additional references.

Reviewer #2, Comment 4:

The Eide group has demonstrated that intrathecally injected contrast is drained to cervical lymph nodes and also makes a plausible argument for dorsal lymphatics. (Ringstad & Eide Nature Communications 2020 among other papers) Kipnis & Reich used IV contrasts of different sizes to visualize the dorsal intracranial lymphatics in monkeys and humans (2017). The Nedergaard lab at University of Rochester showed intrathecal tracer distributing to the eye in mice (2020), Johnston at University of Toronto observed intrathecal tracer surrounding the olfactory nerve and at the lymphatic endothelium structures in the nasal area (2003). This year, basal lymphatic structures in mice were also published by Ahn et al (Nature). The manuscript would benefit from trying to expand the work and novelty factor, perhaps with comparing a control to a patient group, such as MS, where extracranial lymphatics are expected to be increased based on findings from animal models.

Response to Reviewer #2, Comment 4:

We appreciate these thoughtful comments and are encouraged by the reviewer's suggestions. Our study is a retrospective large cohort, and our key objective is the full description of CNS-associated lymphatics in human brains noninvasively. Novelty of our paper is described in the previous two answers. We agree with the reviewer, there is an excellent opportunity to study MS-CNS-associated lymphatics using our technique. We are planning to submit a grant for this purpose in the near future.

Reviewer #2, Comment 5:

The study is also generally quite under-referenced, example: 'Arachnoid granulations and villi of the intracranial and spinal venous sinuses are specialized to filter smaller, acellular particles. Most CSF drains directly into the bloodstream through these structures.' The evidence for this is debatable and needs references. In summary, although the topic is of great interest, the method appears to need more validation in order to make the conclusions presented. The novelty factor could be improved, and more references are needed.

Response to Reviewer #2, Comment 5:

We sincerely thank the reviewer for these encouraging comments regarding the importance of our study and fully acknowledge the reviewer's desire validation of our novel technique. We have now incorporated the important features of recent studies into our manuscript and added an additional 20 references. Also, we discussed an important study (Pollay M. et al., *Cerebrospinal Fluid Res.* 2010, 7:9) for dual CSF absorption in light of the reviewer's comment.

We believe our novel phantom study provides a validation for our technique and the novelty factor of our study is improved with this effort. Our study also has internal validation with age- and sex-associated changes and is consistent with previous research and provides a deep perspective of the current literature.

REVIEWER COMMENTS

Reviewer #1 (Remarks to the Author):

The authors have responded well to the critique. However, they are clearly not familiar with the literature on the meningeal lymphatic literature and this section of the manuscript will need to be rewritten:

1. A better description of dural lymphatics through history:

The authors should highlight that dural lymphatics were not discovered by references (1) and (2). Rather they were described more than 200 years ago by Italian anatomist Paolo Mascagni. Furthermore, even in modern times they have been described in multiple publications using both electron microscopy, histology and immunohistochemistry. See e.g.:

Sandrone, S., Moreno-Zambrano, D., Kipnis, J. & van Gijn, J. A (delayed) history of the brain lymphatic system. *Nat. Med.* 25, 538–540 (2019).

Kida, S., Weller, R. O., Zhang, E. -T, Phillips, M. J. & Iannotti, F. Anatomical pathways for lymphatic drainage of the brain and their pathological significance. *Neuropathol. Appl. Neurobiol.* 21, 181–184 (1995).

Andres, K. H., von Düring, M., Muszynski, K. & Schmidt, R. F. Nerve fibres and their terminals of the dura mater encephali of the rat. *Anat. Embryol. (Berl)*. 175, 289–301 (1987).

2. A more clear distinction between parasagittal spaces and dural lymphatics

The authors do not seem to distinguish between dural lymphatic vessels and parasagittal spaces. This needs to be done as these are separate entities. This is a problem both in the introduction but also when presenting results. E.g. in figure 2 where they described “dural lymphatic signals” from a structure that is likely the parasagittal space. The parasagittal spaces have been shown to drain CSF in humans, and in mice they have been shown to be important interfaces between CSF antigens and immune cells transported by the blood (see Rustenhoven et al, *Cell*, 2021).

Specific comments:

Line 92, page 3. "Human studies also show similar evidence of CSF-ISF drainage through the arachnoid granulations and MLVs."

Please cite the human studies that show drainage through arachnoid granulations and dural lymphatics. To this reviewer's knowledge, the information from humans for either of these pathways is scarce, and mostly post-mortem. Furthermore, while animal studies have shown meningeal lymphatics to carry intrathecally delivered tracers with in vivo imaging, our understanding of the drainage mechanism through arachnoid granulations is almost entirely based on ex vivo, and a few in vitro studies. No convincing in vivo imaging have shown CSF egress through arachnoid granulations. This should be mentioned..

Reviewer #3 (Remarks to the Author):

This manuscript used a novel technique for identifying lymphatic drainage to the deep cervical lymph nodes. Thank you for addressing the comments raised. While the technique is impressive and could be very useful in showing the perineurial and periarterial pathways for lymphatic drainage into the cervical lymph nodes, I have some major concerns.

1. The authors are very dismissive of IPAD mainly because they cannot visualise it using this technique. The composition of ISF within the extracellular spaces of the brain is exactly the same as that within the basement membranes of the intracerebral capillaries and arteries (the IPAD pathways), so how would a difference in signal intensity be seen if the authors themselves state that the difference in signal intensity is due to the different composition of fluid. Furthermore, authors state in their response that the technique described here "cannot visualise lymphatic tissue itself but rather can detect lymphatic fluid ...due to sensitivity of this technique to elevated protein concentration" . Since the fluid in the intracerebral IPAD pathways is exactly the same fluid as in the extra cerebral spaces of the cerebral parenchyma, it is impossible for this technique to detect it in the IPAD pathways. IPAD may not be seen using this technique because the signal intensity within the intracerebral IPAD pathways is as it should be- exactly the same as that in the extracellular spaces of the brain. Besides, the IPAD pathways are small and inaccessible, so they cannot possibly be resolved by the technique employed here. This requires acknowledgement, rather than a dismissal that IPAD does not exist and the "lymphatic theory is the most accepted theory to date";

2. Thank you for adding references that are extremely relevant to the work. It is very obvious that the authors are using up to date references only for the lymphatic interpretations while the IPAD references are over a decade old- in fact the paper where the lymphatic term was first mentioned Albargothy et al (Acta Neuropathologica 2016) is not even mentioned. I suggest replacing some of the references with newer ones.

3. In response to reviewer 2 of whether there was a cut off value used, the authors state there was no cut-off value due to the wide variation between anatomical structures- how is the control then ensured?

Reviewer #4 (Remarks to the Author):

The authors describe a non invasive method for assessing lymphatic channels in around the brain and meninges. The paper is timely since there is increasing interest in visualising the cerebral lymphatic system in humans in vivo. I found some points lacked detail and that could be improved.

General: lymphatics in the brain including those draining through the cribriform plate and including in humans were described decades ago and 'rediscovered' more recently - please correct some inaccurate statements about the history. (eg remove 'putative'). You suggest in the introduction that the arachnoid granulations pass fluid to the lymphatics but I don't think this is correct - fluid passes through arachnoid granulations into the venous sinuses.

The phantom data are useful. How old was the healthy human subject that was scanned simultaneously? How did the authors deal with the problems of variation in field along the bore of the coil to avoid signal distortion? The conc of albumin was varied - what else was in the solution to help mimic interstitial fluid or CSF? CSF is not just water with some protein. You should at least discuss the effect of other solutes and debris in the ISF/CSF that you have not accounted for.

The study uses patient brain scans that included a 3D FLAIR sequence performed for various reasons in persons attending the authors department. Described as retrospective.

1. Please indicate the main reasons for the scans being performed ie what was the disease indication, including the number and proportion of patients in each category.
2. Please provide a consort diagram indicating the total number of cases attending the department within the time period and reasons for exclusion, so the reader can understand how the cases were selected and the generalisability of the results.
3. Was the FLAIR sequence isotropic? If not was it manipulated to become isotropic?

The lymphatic identification and quantification needs more detail - how was the 'signal' and 'maximum thickness' measured? These were presumably obtained at standard locations or

anatomical points - please describe in more details - this is a crucial part of your analysis but at the moment there is no detail at all about how this was done.

What was the reliability of the measurements - did your single rater perform repeated assessments to quantify intra-rater reliability?

How was the rater blinded to any brain pathology or patient age or sex knowledge of which might have influenced their measurements?

FLAIR signal is not standard, absolute or quantitative, but varies with scanner settings and a range of patient and other factors - how did you standardise the signal settings so as to ensure that your lymphatic signal readings were 'standard'? A diagram of how the measurements were obtained, and how they were combined to give dorsal and ventral values would help.

Some statements need referencing. eg line 233: 'Age is associated with reduction in muscle cell mass within the lymphatic vascular wall, resulting in contractile dysfunction and enlarged lymphatic diameter. Aging is a major risk factor for decreased pump activity in the lymphatic system.' - what is the evidence for this in cerebral lymphatics, or are you extrapolating from systemic lymphatics? How can you differentiate impaired systemic lymphatic function due to loss of striatal muscle activity from issues due to loss of contractile cells in the lymphatic channels? Is this from the Shang reference? Either way, such statements should be referenced and specific.

Statistics:

It would be essential to account for age and gender and potentially also head/intracranial/brain size in your analysis - apologies if I missed this but I did not see any covariate adjusted analyses.

The sensitivity/specificity and ROC curve approach to assess age does not seem an appropriate analysis. You do not have a reference standard so how can you assess sensitivity and specificity? You should perform a co-variate adjusted regression model or similar approach?

Lymphatic size may vary with brain or head size - you should account for this (see below). This alone may account for apparent male-female differences.

Results: The Table of measures of the various channels should be in the main paper not the suppl.

The statements in methods about lymphatics becoming larger with age seem to contradict your own findings where more lymphatics were more often visible in younger than older subjects.

The additional problem in the cranial cavity is that as the brain shrinks with age, the other structures such as the venous sinuses increase in size to take up the vacated space - perhaps this also happens to the lymphatics - this is why you need to consider some other covariates in your analyses.

The data on lymphatics around the major arteries is not very convincing. What about the vasa vasorum? How do you differentiate periarterial lymphatics from vasa vasorum? I think you are just seeing the ICA wall and not lymphatic channels - can you provide more convincing data? Just because the 'perivascular channels' become harder to see below the cervical lymph nodes does not mean that the structures above the lymph nodes are lymphatic channels - it could just be the effect of signal drop out at the lower limit of the MRI coil.

I would be very careful about citing the Mestre work on CSF uptake into perivascular spaces in acute stroke as being a major source of brain oedema - unfortunately this theory fails to account for the well documented changes in humans of reduced ADC indicating increased intracellular oedema. SUGest this is not relevant to your current work and could be omitted.

There are other recent publications in visualisation of meningeal lymphatics which are worth citing - eg Ding et al Nature Med 2021;27:411-418.

I did not find the videos at all helpful - these could be dropped.

Responses to the Reviewers' Comments:

We sincerely appreciate the reviewers' careful analysis and constructive suggestions to improve the quality of our manuscript, and we hope that these revisions will now make our work suitable for publication.

Reviewer 1 indicated that "*The authors have responded well to the critique...*" but raised questions about the description of the literature.

Reviewer 3 pointed out that "*technique is impressive*" and "*could be very useful in showing the perineurial and periarterial pathways for lymphatic drainage*" but raised concerns about the methodology and its functional readouts.

Reviewer 4 highlighted that "*The paper is timely since there is increasing interest in visualizing the cerebral lymphatic system in humans in vivo*" but raised questions about the structure of the manuscript and the methodology.

Reviewer #1, Comment 1:

The authors should highlight that Dural lymphatics were not discovered by references (1) and (2). Rather they were described more than 200 years ago by Italian anatomist Paolo Mascagni. Furthermore, even in modern times they have been described in multiple publications using both electron microscopy, histology, and immunohistochemistry. See e.g.:

Sandrone, S., Moreno-Zambrano, D., Kipnis, J. & van Gijn, J. (2019) A (delayed) history of the brain lymphatic system. *Nature Med.* 25:538–540.

Kida, S., Weller, R. O., Zhang, E. T., Phillips, M. J. & Iannotti, F. (1995) Anatomical pathways for lymphatic drainage of the brain and their pathological significance. *Neuropathol. Appl. Neurobiol.* 21(3):181-4.

Andres, K. H., von Düring, M., Muszynski, K. & Schmidt, R. F. (1987) Nerve fibres and their terminals of the dura mater encephali of the rat. *Anat. Embryol. (Berl).* 175(3):289-301.

Response to Reviewer #1, Comment 1:

We are grateful to the reviewer for bringing up this important point and apologize for our oversight. We have now rewritten the related introduction paragraph (second paragraph–first 4 lines), included Mascagni's study, and added three new references about the history of dural lymphatics.

Reviewer #1, Comment 2:

The authors do not seem to distinguish between Dural lymphatic vessels and parasagittal spaces. This needs to be done as these are separate entities. This is a problem both in the introduction but also when presenting results. E.g., in figure 2 where they described Dural lymphatic signals from a structure that is likely the parasagittal space. The parasagittal spaces have been shown to drain CSF in humans, and in mice they have been shown to be important interfaces between CSF antigens and immune cells transported by the blood (see Rustenhoven et al, *Cell*, 2021).

Response to Reviewer #1, Comment 2:

We fully acknowledge the reviewer's concerns and apologize for not carefully distinguishing between dural lymphatic vessels and parasagittal spaces. The parasagittal dura region has been identified earlier by Fox et al., *Neurosurgery*, 1996 and Han et al., *Neuroradiology*, 2007. Furthermore, Rinstad et al., *Nature Communications*, 2020 described escape of contrast material from cerebrospinal fluid (CSF) into parasagittal dura along the superior sagittal sinus at areas nearby entry of cortical cerebral veins. In addition, similar parasagittal CSF antigen efflux was observed in a recent animal study (Rustenhoven et al., *Cell*, 2021), and was emphasized as by the term "*perisinus dura*," describing the site at which CSF antigens first encounter sinus-associated antigen presenting cells and that subsequently drains via meningeal lymphatics. We have adjusted all terminology referring to dural lymphatics to terms of *parasagittal* or *perisinus* lymphatics (for transverse, sigmoid, sinus rectus etc.) in the introduction, results, discussion, methods, and figure legends, particularly in the Fig 2.

Reviewer #1, Specific comments:

Line 92, page 3. Human studies also show similar evidence of CSF-ISF drainage through the arachnoid granulations and MLVs. Please cite the human studies that show drainage through arachnoid granulations and Dural lymphatics. To this reviewer's knowledge, the information from humans for either of these pathways is scarce, and mostly post-mortem. Furthermore, while animal studies have shown meningeal lymphatics to carry intrathecally delivered tracers with *in vivo* imaging, our understanding of the drainage mechanism through arachnoid granulations is almost entirely based on *ex vivo*, and a few *in vitro* studies. No convincing *in vivo* imaging have shown CSF egress through arachnoid granulations. This should be mentioned.

Response to Reviewer #1, Specific comments:

We fully agree with the reviewer and apologize for our oversight that there is no *in vivo* imaging data having shown CSF drainage through arachnoid granulations. We have rewritten our introduction (first paragraph, line 7-11) based on the suggestion of the reviewer.

Reviewer #3, Comment 1:

This manuscript used a novel technique for identifying lymphatic drainage to the deep cervical lymph nodes. Thank you for addressing the comments raised. While the technique is impressive and could be very useful in showing the perineurial and periarterial pathways for lymphatic drainage into the cervical lymph nodes, I have some major concerns.

The authors state that this is the first to describe how FLAIR MR can be used for detection of lymphatic structures. It is an issue that the sequence is used both presented as novel method without extensive validation using established methods, however, is simultaneously used to identify new or under-appreciated cervical lymphatic structures. The authors are very dismissive of iPAD mainly because they cannot visualize it using this technique. The composition of ISF within the extracellular spaces of the brain is exactly the same as that within the basement membranes of the intracerebral capillaries and arteries (the IPAD pathways), so how would a difference in signal intensity be seen if the authors themselves state that the difference in signal intensity is due to the different composition of fluid. Furthermore, authors state in their response that the technique described here "cannot visualize lymphatic tissue itself but rather can detect lymphatic fluid ...due to sensitivity of this technique to elevated protein concentration". Since the fluid in the intracerebral IPAD pathways is exactly the same fluid as in the extra cerebral spaces of the cerebral parenchyma, it is impossible for this technique to detect it in the IPAD pathways. iPAD may not be seen using this technique because the signal intensity within the intracerebral iPAD pathways is as it should be exactly the same as that in the extracellular spaces of the brain. Besides, the IPAD pathways are small and inaccessible, so they cannot possibly be resolved by the technique employed here. This requires acknowledgement, rather than a dismissal that IPAD does not exist and the "lymphatic theory is the most accepted theory to date".

Response to Reviewer #3, Comment 1:

We fully agree that our arterial findings are a bit confusing and cause conflict between the glymphatic theory and iPAD theory. In our study, we detected prominent signals around the petrous internal carotid artery (ICA) at the level of the skull base but not in intracranial arterial structures including supraclinoid ICAs. There is no continuation from the intracranial arterial wall to the skull base. The signal appears at the level of the skull base and around the petrous internal carotid artery. The etiology or source of the signal is not clear at this point. We can see this type of the signal with the T1 SPACE post-gadolinium series, as seen below in our clinic patients. T1 SPACE post-gadolinium is very sensitive for evaluation of the vessel wall (*images provided below*). The signal appears at the level of the skull base, with no significant signal seen in the supraclinoid ICA or below the lymph nodes. We assume that signal might be related to the sympathetic plexus around the ICA and possibly a small pathway between the skull base and the lymph nodes. We fully agree that this finding cannot be fully explained and may cause confusion for the reader about iPAD theory vs Glymphatic theory. Data regarding our arterial

findings may not yet be mature, though these findings are only a small part of this study. We therefore removed our arterial findings from this study due to this possibility for misperception. We rephrased our manuscripts and detached arterial figures and related information. We are planning a comprehensive study for these arterial findings in the future.

Reviewer #3, Comment 2:

Thank you for adding references that are extremely relevant to the work. It is very obvious that the authors are using up to date references only for the lymphatic interpretations while the iPad references are over a decade old- in fact the paper where the lymphatic term was first mentioned Albargothy et al (*Acta Neuropathologica* 2016) is not even mentioned. I suggest replacing some of the references with newer ones.

Response to Reviewer #3, Comment 2:

We fully agree with the reviewer that the using up to date references would strengthen the manuscript. We included a few recent important papers that support our data, including Mezey et al., *PNAS* 2021; Yagmurlu et al., *Brain Sci* 2020; and Giannetto et al., *Sci Reports* 2020, Albargothy et al (*Acta Neuropathologica*).

According to the reviewer's first comment, we rephrased iPad related information and detached the arterial data including the references.

Reviewer #3, Comment 3:

In response to reviewer 2 of whether there was a cut off value used, the authors state there was no cut-off value due to the wide variation between anatomical structures- how is the control then ensured?

Response to Reviewer #3, Comment 3:

We fully acknowledge the reviewer's desire for more explanation regarding cut-off values. In our study, we have two main goals. The first goal was non-invasive visualization of the ventral and dorsal lymphatic structures and their connections with the cervical lymph nodes. Our study population included only seizure subjects without lesions. Previous human studies with intravenous or intrathecal contrast infusion revealed parasagittal lymphatic signals along the sagittal sinus or anterior cranial fossa. These studies identified the lymphatic structures by contrast enhancement or increased signal intensity in those areas relative to adjacent sinuses or CSF spaces. All those human studies evaluated qualitatively, especially for those with intravenous contrast administration. In order to identify the dorsal and ventral lymphatic structures, we used 3D-T2-FLAIR which is uniquely sensitive to the

molecular content of fluids. We can differentiate the lymphatic structures from CSF spaces and the sinus lumen, since FLAIR sequences can suppress signals derived from venous flow and normal CSF. Signals with the highest intensity around the parasagittal sinuses are accepted as lymphatic signals, based on findings recently described in the human and animal studies. We measured this parasagittal signal intensity and thickness within key target regions. Within ventral lymphatic structures, we identified the highest signal intensities around the cranial nerves, calvarial foramina and anterior cranial fossa in the dural regions. We would like to emphasize that the first part of this study is observational and descriptive. Using a cut off value to determine visibility was not feasible in this study. When the signal intensity at targeted anatomical location of a lymphatic vessel was not detectable, we acknowledged as not having a visible lymphatic vessel.

Alternatively, when there was a detectable signal intensity at a specific region of interest compatible with lymphatic vessels, thickness and signal intensity values were measured. When comparing visibility between groups, we used the chi-square test. When comparing median thickness and mean SI values, we excluded each case that did not possess a visible vessel at that region. So, as an example, when we compared transverse sinus thickness and signal intensity values, comparison was performed among 57 cases with visible transverse sinus lymphatics instead of 81 cases.

The second goal of the study was to assess age-related changes in the brain lymphatics system. Here, we used a cut off value for the differentiation of young and elderly, which was 50 years of age. The age of 50 years discriminated well between lower and higher than mean dorsal (area under the curve 0.68, 95% CI 0.56-0.79, $p=0.007$, sensitivity 50%, and specificity 76.3%) and ventral SI/thickness ratios (area under the curve 0.73, 95% CI 0.62-0.85, $p<0.001$, sensitivity 58%, and specificity 85.4%). Binary regression analyses revealed that an age older than 50 was associated with lower dorsal and ventral SI/thickness ratios, independent of sex and intracranial volume measurements (OR 2.9, 95% CI 1.1-7.7, $p=0.029$ and OR 8.3, 95% CI 2.8-24.7, $p<0.001$, respectively). We included this new information in the Method and Result sections.

Reviewer #4, General Comments and Responses:

The authors describe a non-invasive method for assessing lymphatic channels in the brain and meninges. The paper is timely since there is increasing interest in visualizing the cerebral lymphatic system in humans *in vivo*. I found some points lacked detail and that could be improved.

Reviewer #4, General Comment 1:

Lymphatics in the brain including those draining through the cribriform plate and including in humans were described decades ago and 'rediscovered' more recently - please correct some inaccurate statements about the history. (eg remove 'putative'). You suggest in the introduction that the arachnoid granulations pass fluid to the lymphatics, but I don't think this is correct - fluid passes through arachnoid granulations into the venous sinuses.

Response to Reviewer #4, General Comment 1:

We are grateful to the reviewer for bringing up these important points and apologize for our oversight. We have now rewritten the related introduction paragraph (Second paragraph-first 4 line) included Mascagni's study and added three new references about the history of dural lymphatics (Mezey et al., *PNAS* 2021; Yagmurlu et al., *Brain Sci* 2020; Giannetto et al., *Sci Reports* 2020). We have also removed the statement of "putative" from the manuscript and figure legends.

We fully acknowledge the reviewer's concerns and apologize for our oversight that CSF drainage into the venous sinuses and CSF-ISF drainage via meningeal lymphatics are separate. Accordingly, we have rewritten our introduction (First paragraph-line 5-11) based on the suggestion of the reviewer.

Reviewer #4, General Comment 2:

The phantom data are useful.

How old was the healthy human subject that was scanned simultaneously?

Response to Reviewer #4, General Comment 2:

The healthy human subject was a 48-year-old male. In response to the reviewer's comment, we added this information into the method section - MRI Protocol: *Phantom imaging and Phantom MR imaging with a healthy human subject* (Paragraph 4 line 1, 2) and the legend of Figure 1.

Reviewer #4, General Comment 3:

How did the authors deal with the problems of variation in field along the bore of the coil to avoid signal distortion?

Response to Reviewer #4, General Comment 3:

We thank the reviewers for raising this important point regarding possible problems of variation in field along the bore of the coil to avoid signal distortion. We have faced this problem during our pilot experiments. Our first set of experiments showed multiple artifacts with signal distortions. We resolved this issue by shimming before scanning and through changing the epicenter of the scan to the center portion of the phantom. The images from the phantom studies are given below, including scans made (A) before the shimming, and epicenter correction, and (B) after the shimming and epicenter correction.

Reviewer #4, General Comment 4:

The conc of albumin was varied - what else was in the solution to help mimic interstitial fluid or CSF? CSF is not just water with some protein. You should at least discuss the effect of other solutes and debris in the ISF/CSF that you have not accounted for.

Response to Reviewer #4, General Comment 4:

We fully agree with the reviewer that the lymphatic fluid does not consist only of albumin, but also includes cells, fragments of the cells, solutes, and debris. However, as we know from the previous studies, albumin is the most abundant protein in the lymph, contributing to 65% of the total protein content of lymph in the body (Rutili et al., *Acta Physiol. Scand* 1977), and the albumin concentration in the interstitial fluid (ISF) is up to 30% of that in the plasma (Ellmerer et al., *Am J Physiol Endocrinol Metab.* 2000). Notably, the albumin concentration in interstitial fluid is very similar to its concentration in lymph (~1.6–1.8 g/dl) (Rutili et al., *Acta Physiol. Scand* 1977; Abdallah et al., *J of Controlled Release* 2020), which is consistent with our phantom study's albumin concentration formulated to emulate the lymphatic signals from the healthy subject (between 2-4 g/dl). Our measurements are

therefore higher than those observed in systemic lymphatic fluid, which might be related to additional components of *in vivo* lymphatic fluid, such as cells, solutes, debris, and other proteins. Furthermore, formulations comprising albumin as a single solute have been used to evoke mock lymphatic signals using a number of imaging techniques, including MRI and PET (Wang et al., *PNAS* 2015; Ruddell et al., *J. Magn. Reson. Imaging* 2015).

Unfortunately, therefore, it may not be feasible to perform phantom studies with natural lymphatic fluids. Nonetheless, we acknowledged the reviewer's concerns and designed another phantom study using commercially available synthetic lymphatic-like fluid material to validate our technique. Only one simulated lymphatic fluid it was not plausible to find a significant amount of lymphatic fluid from a real patient or a healthy subject. Furthermore, the components of the lymphatic fluid are not the same for all organs and tissues and might change due to tissue degeneration, inflammation, metabolism, and the stages of anabolism or catabolism etc. commercially available, (BioCHEMAZONE, Simulated Lymph Fluid; <https://biochemazone.com/product/artificial-lymph-fluid/>) which has not been validated to represent lymphatic fluid as measured *in vivo* in MRI studies. We scanned this artificial lymphatic fluid. In this experiment, however, we detected significant signal loss in the artificial lymphatic fluid in our fat saturated 3D FLAIR images. We suspect that the company formulated this artificial lymphatic fluid to simulate that of the main thoracic duct, which includes significant fat from the gastrointestinal system. We inquired about the ingredients of this company's artificial lymphatic fluid, but the company does not share this proprietary information due to their confidentiality policy. Therefore, it was not feasible to use this artificial fluid for our phantom study. We added a paragraph about the limitations of this phantom study and plausible effects of other components of the interstitial fluid and *in vivo* lymphatic fluid per se in the result section, MR phantom study (Paragraph 2 line 10, 19) and in the discussion-limitations.

Reviewer #4, General Comment 5:

The study uses patient brain scans that included a 3D FLAIR sequence performed for various reasons in persons attending the authors department. Described as retrospective.

Please indicate the main reasons for the scans being performed ie what was the disease indication, including the number and proportion of patients in each category.

Response to Reviewer #4, General Comment 5:

We appreciate the reviewer's comment. The indication of MR studies was seizure. All clinical MR studies were done for the seizure. This was described in the Methods section, *Experimental Design and Study Subjects*, Line 2. We have a chart that shows the patient selection criteria and the flow chart, Supplement Fig 1.

Reviewer #4, General Comment 6:

Please provide a consort diagram indicating the total number of cases attending the department within the time period and reasons for exclusion, so the reader can understand how the cases were selected and the generalisability of the results.

Response to Reviewer #4, General Comment 6:

We apologize for our oversight that our flow chart can be seen in the Figure supplement 1, which includes the total number of cases attending the department within the time period and reasons for exclusion.

Reviewer #4, General Comment 7:

Was the FLAIR sequence isotropic? If not, was it manipulated to become isotropic?

Response to Reviewer #4, General Comment 7:

We thank the reviewers for raising this important point. It was isotropic allowing use MPR, which can evaluate anatomical structures in three planes. We included this information in the Method section- *Patient's MR imaging* - Line 3 and *Clinical MR Imaging* - Line 2.

Reviewer #4, General Comment 8:

The lymphatic identification and quantification need more detail - how was the 'signal' and 'maximum thickness' measured? These were presumably obtained at standard locations or anatomical points - please

describe in more details - this is a crucial part of your analysis but at the moment there is no detail at all about how this was done.

Response to Reviewer #4, General Comment 8:

We fully acknowledge the reviewer's desire for more detailed identification and description for our analysis. The meningeal lymphatic signal has recently been shown with intravenous or intrathecal injection of gadolinium-based contrast agents by using 3D-FLAIR MR imaging (Absinta M. et al., *Elife* 2017; Ringstad G. et al., *Nat Commun.* 2020; and Zhou Y. et al., *Ann Neurol.* 2020). The meningeal lymphatic signal runs alongside the sagittal sinus in those human MR studies, especially in the mid portion of the superior sagittal sinus. Related illustrations are included below from each of these prior publications, respectively.

In our study, we detected significant signal intensity on 3D isotropic FLAIR series without contrast administration in the locations of lymphatic structures described in previous human MR studies along the sagittal sinus and anterior cranial fossa in all 81 human subjects studied. The lymphatic signal along the sinuses in the dorsal system is seen below.

We know that this lymphatic flow drains to cervical lymph nodes based on animal and human studies (Ahn et al., *Nature* 2020; Eide et al., *Scientific Reports* 2018). Therefore, we emphasized that the lymphatic signals would be around all intracranial venous structures and should be continuous along the venous pathway until reaching the cervical lymph nodes in humans. Indeed, a recent autopsy study proved this by demonstrating lymphatic vessels connecting directly to lymph nodes adjacent to the internal jugular vein within the carotid sheath in the upper neck region (Yagmurlu et al., *Brain Sci* 2020). The majority of the lymphatic vessels drain to the retropharyngeal lymph nodes. We decided to look at all intracranial venous sinuses and dural structures. Of note, we observed that dural venous sinuses and CSF were both dark in our 3D-T2-FLAIR sequence parameters, whereas meningeal lymphatic structures were seen as bright linear areas surrounding the venous sinuses or central dural structures in 3D-T2-FLAIR in the dorsal surfaces of the brain.

Our non-contrast 3D T2-FLAIR MR imaging parameters permitting detailed visualization of dorsal putative meningeal lymphatic vessels (adapted from the Figure 2).

Representative meningeal lymphatic vessels (mLVs) were proficiently drawn by the neuroradiologist via single point ROI. The neuroradiologist always found the maximal intensity region in the mLVs regions using multiplanar reconstruction of the 3D-T2-FLAIR images. Later, the neuroradiologist measured the maximal thickness of mLVs again using multiplanar reconstruction of the 3D-T2-FLAIR images. Maximum thickness and maximum signal intensity (SI) were measured along the anterior, middle, and posterior sagittal sinus, sinus rectus, confluence, bilateral transverse sinus, bilateral transverse sinus-sigmoid sinus junction, bilateral sigmoid sinuses, bilateral jugular veins, and posterior aspect of the foramen magnum. Superior sagittal sinus divided into three segments, including anterior segments before the coronal suture, middle segment between coronal and lambdoid sutures, and the posterior segments between the lambdoid suture and confluence. These structures are accepted as the dorsal lymphatic system. Each anatomical structure was measured in the projection that showed the greatest max signal intensity and perpendicular to the maximum thickness. For example, superior sagittal sinuses are generally measured on the coronal series, confluences on sagittal or coronal series, and transverse sinus-sigmoid sinus junction on the axial or coronal series. We did not necessarily always acquire paired SI and thickness values for a given ROI in each patient from the same individual image slice from that patient. In animal studies, ventral lymphatic pathways are much more pronounced relative to the dorsal pathways, but in humans, this ventral lymphatic flow can be reliably imaged only along the dura and olfactory nerve in the anterior cranial fossa. Previous human studies using heavily T2-weighted fluid-attenuated inversion recovery (hT2w-FLAIR) MRI revealed that inner ear structures, optic nerves, and Meckel caves show significant gadolinium absorption, based on measurements obtained before, and 3 hours and 24 hours after intravenous administration of gadolinium-based contrast agent (GBCA) (Deike-Hoffman et al., *Invest Radiol* 2019). We noticed a similar pattern of signal enhancement in the ventral dura around the major skull base foramina without gadolinium injection in our 3D-T2-FLAIR sequence parameters. Specifically, we managed to visualize increased dural signal intensity around the major cranial nerves, anterior cranial fossa dura, olfactory groove adjacent to the optic nerves, around the Meckel cave orifices, in the Meckel cave, around the internal auditory canal, and within the jugular foramen. These signals might represent the major components of the ventral lymphatic system in the human brain. Again, dural blood vessels and CSF were both dark in our 3D-T2-FLAIR sequences, whereas mLVs were white linear areas surrounding the dura at the level of the major cranial foramina. Maximum thickness and maximum SI were also measured in the dural/paradural structures along the anterior cranial fossa, optic groove, diaphragma sellae, bilateral dura at the orifice of the Meckel caves, bilateral dura in the Meckel caves, bilateral dura at the orifice of the internal auditory canals, and bilateral dura at the level of the orifices of jugular foramen. These structures are accepted as the ventral lymphatic system. A few representative approaches to measurement are shown below. Each anatomical structure is measured in the projections that depicts the maximum signal intensity

and perpendicular to the maximum thickness as detailed below. We did not necessarily always acquire paired SI and thickness values for a given ROI in each patient from the same individual image slice from that patient. We improved the description our lymphatic identification and quantification method according to the approach shown below and described how the 'signal' and 'maximum thickness' were measured. We also rewrote the methods section to reflect these changes.

The max signal intensity (SI) (A) and thickness (B) measurements of the transverse-sigmoid junction on the right are seen in three projections (Subject 24).

The max signal intensity (SI) (A) and thickness (B) measurements of the dura of the anterior cranial fossa dura around the olfactory nerves are seen in three projections (Subject 24). (adapted from Supplementary Figure 5).

The max signal intensity (SI) (A) and thickness (B) measurements of dura at the level of the orifice of the Meckel's cave around the trigeminal nerves are seen in three projections (Subject 41).

Reviewer #4, General Comment 9:

What was the reliability of the measurements - did your single rater perform repeated assessments to quantify intra-rater reliability?

Response to Reviewer #4, General Comment 9:

We appreciate the reviewer for this relevant and helpful comment. For the reliability analyses, the signal intensity and thickness of the first 10 subjects were evaluated. Notably, an intra-class correlation (ICC) coefficient analysis was performed to determine intra-rater reliability and revealed a significant reliability. The ICC values were near 1 in all regions analyzed, representing high correlation in measurements independently collected. We added this information to the *Methods* and *Results* sections accordingly.

Reviewer #4, General Comment 10:

How was the rater blinded to any brain pathology or patient age or sex knowledge of which might have influenced their measurements?

Response to Reviewer #4, General Comment 10:

We fully acknowledge the reviewer's desire for reliability of our study. All patients had been de-identified including, name, sex, and age first. The reader was blinded from all identifications including age, sex, name, date

of birth during the measurements, statistical analyses, and evaluations. These details are given in the Method section - *Image Analysis* and *Clinical MR Imaging*, Line 4.

Reviewer #4, General Comment 11:

FLAIR signal is not standard, absolute or quantitative, but varies with scanner settings and a range of patients and other factors - how did you standardize the signal settings so as to ensure that your lymphatic signal readings were 'standard'? A diagram of how the measurements were obtained, and how they were combined to give dorsal and ventral values would help.

Response to Reviewer #4, General Comment 11:

We are grateful to the reviewer for bringing up this important point. We fully agree that the FLAIR signal might change with scanner technique/setting, and a range of patient or other factors. We used the same single scanner (3T Prisma machine) and the same scanner settings (TR, TE, IR times and other sequence parameters) in each patient studied. We did not use raw FLAIR signal intensity (SI) values for statistical analyses, including in comparisons relating to age and sex. We sometimes used central white matter signal intensity for normalization, and sometimes used muscle of the superior temporal muscle signal. Age and sex related comparisons were calculated with internal normalization with muscle or white matter. Although we used the muscle/lymphatics SI ratio for age, we did not use muscle-lymphatics SI ratio for sex due to sex difference for the muscle signals. Consequently, sex related comparison was calculated with white matter-lymphatics SI ratio for healthy comparison. SI of the reference areas including white matter, gray matter, and muscle and their association with age and sex are provided in Supplementary Table 4. We acknowledged these details in the Supplementary Tables 2, 3 and 4. We also included internal normalization information for age and sex comparisons in the Method and Result sections.

Furthermore, we included a diagram and images for our descriptive measurement techniques in the supplemental section for dorsal and ventral regions (Supplementary Figure 5). We also include our ventral and dorsal combined list in this figure. Dorsal and ventral variables that could be seen in all patients were the only variables used in statistical analysis (Supplementary Figure 5).

Reviewer #4, General Comment 12:

Some statements need referencing. eg line 233: 'Age is associated with reduction in muscle cell mass within the lymphatic vascular wall, resulting in contractile dysfunction and enlarged lymphatic diameter. Aging is a major risk factor for decreased pump activity in the lymphatic system.' - what is the evidence for this in cerebral lymphatics, or are you extrapolating from systemic lymphatics? How can you differentiate impaired systemic lymphatic function due to loss of striatal muscle activity from issues due to loss of contractile cells in the lymphatic channels? Is this from the Shang reference? Either way, such statements should be referenced and specific.

Response to Reviewer #4, General Comment 12:

We are grateful to the reviewer for bringing up this important point and apologize for our oversight. Our intent was to report the muscle cell atrophy in the wall of the lymphatic vessels, destruction of elastic elements, and alterations in the lymphatic contractility with aging, not skeletal muscles in the extremities. We are aware that Shang et al. reported decreased skeletal-striated muscle activity and atrophy as a reason for the decreased lymphatic output in the body (Shang et al., *Aging* 2019), which is not an issue in the brain. We rephrased related paragraphs in the Methods section with new references.

Reviewer #4, Statistics Comment 1:

It would be essential to account for age and gender and potentially also head/intracranial/brain size in your analysis - apologies if I missed this but I did not see any covariate adjusted analyses.

Response to Reviewer #4, Statistics Comment 1:

We thank the reviewer for this practical and positive comment. We performed partial correlation analysis to see if the correlations were valid when adjustments for age, sex, and intracranial volume were performed. Notably,

there was no significant change after adjusting for these variables. We added this information to the Methods and Results sections.

Reviewer #4, Statistics Comment 2:

The sensitivity/specificity and ROC curve approach to assess age does not seem an appropriate analysis.

Response to Reviewer #4, Statistics Comment 2:

We thank the reviewer for this relevant comment. We did multivariate logistic regression analyses to evaluate if an age cut off of 50 performed well. We found that an age over 50 was associated with significantly lower dorsal and ventral signal intensity/thickness ratios, independent of sex and intracranial volume measurements. We added this information into the Methods and Results sections.

Reviewer #4, Statistics Comment 3:

Lymphatic size may vary with brain or head size - you should account for this (see below). This alone may account for apparent male-female differences.

Response to Reviewer #4, Statistics Comment 3:

We appreciate this invitation to clarify. In order to address this relevant query, we measured the volumes of gray matter, white matter, CSF, and total brain parenchyma (white matter plus gray matter). We analyzed if these parameters were correlated with overall mean dorsal SI and thickness and overall mean ventral signal intensity (SI) and thickness values. Of note, brain volumes were not correlated with these SI and thickness values. There was a significant correlation only between CSF volume and these parameters. The CSF volume was positively correlated with overall mean dorsal and ventral thickness and overall mean dorsal SI, but not with overall mean ventral SI. We also performed partial correlation analyses as mentioned above by adjusting for age, sex, and cranial volume. Of note, age was correlated with dorsal thickness and SI values when adjusted for sex and brain volume (gray and white matter), but not with ventral thickness or SI values. These follow up analyses were added to the Methods and Results sections.

Reviewer #4, Results Comment 1:

The Table of measures of the various channels should be in the main paper not the supplement.

Response to Reviewer #4, Results Comment 1:

We included the main channels in Figures 2 and 3 in the bottom part of the figures for both the dorsal and the ventral systems. We sincerely apologize that we could not include more tables in the main text due to the limitation of table and figure numbers for this journal.

Reviewer #4, Results Comment 2:

The statements in methods about lymphatics becoming larger with age seem to contradict your own findings where more lymphatics were more often visible in younger than older subjects.

Response to Reviewer #4, Results Comment 2:

We are grateful to the reviewer for bringing up this point and apologize for our oversight. We believe that this issue was caused by the color of the pie chart at Figure 6c. Here in this figure, white was 'visible', and gray was 'invisible.' At the same time, the bottom tables-bars show visibility increases with age at Figure 6c. Therefore, to make interpretation of these charts more intuitive, we have modified the legends and charts of this figure such that gray is positive and white is negative. Additionally, we added on a small paragraph in the Result section - *Visibility, Description and Comparison of the Dorsal and Ventral Lymphatic Systems section* - First paragraph-new line 11-14.

Reviewer #4, Results Comment 3:

The additional problem in the cranial cavity is that as the brain shrinks with age, the other structures such as the venous sinuses increase in size to take up the vacated space - perhaps this also happens to the lymphatics - this is why you need to consider some other covariates in your analyses.

Response to Reviewer #4, Results Comment 3:

We thank the reviewer for this relevant comment. We calculated intracranial volume, total neural parenchyma, grey matter, white matter, and CSF volume for each patient using software. We also performed multiple covariates analysis for these variables. We included a new sub-section into the Method section, namely *Volumetric analysis for Intracranial cavity, Tissue Volumes and CSF Volumes* and discussed our findings in the Result section - *Aging-associated changes in brain lymphatic system* (the last paragraph 2 of the sub-section).

Reviewer #4, Results Comment 4:

The data on lymphatics around the major arteries is not very convincing. What about the vasa vasorum? How do you differentiate periarterial lymphatics from vasa vasorum? I think you are just seeing the ICA wall and not lymphatic channels - can you provide more convincing data? Just because the 'perivascular channels' become harder to see below the cervical lymph nodes does not mean that the structures above the lymph nodes are lymphatic channels - it could just be the effect of signal drop out at the lower limit of the MRI coil. I would be very careful about citing the Mestre work on CSF uptake into perivascular spaces in acute stroke as being a major source of brain oedema - unfortunately this theory fails to account for the well documented changes in humans of reduced ADC indicating increased intracellular oedema. Suggest this is not relevant to your current work and could be omitted.

Response to Reviewer #4, Results Comment 4:

We fully agree that our arterial findings are a bit confusing and may cause conflict between the glymphatic theory and iPad theory. In our study, we detected prominent signals around the petrous internal carotid artery (ICA) at the level of the skull base but not in intracranial arterial structures including supraclinoid ICAs. There is no continuation in these structures from the superior arterial wall to the skull base; the signal appears at the level of the skull base and around the petrous internal carotid artery. The etiology or source of the signal is not clear at this point. We can see this type of the signal with the T1 SPACE post-gadolinium series which can be seen below in our clinic patients.

T1-post gadolinium SPACE is very sensitive for evaluation of the vessel wall. Both images are provided below. The signal appears at the level of the skull base, but no significant signal is seen in the supraclinoid ICA or below the lymph nodes. If this is a vasa vasorum related signal, it should start from supraclinoid ICA and progressively

thickens towards the CCA bifurcation or ICA bulbs: a pattern we cannot see. We assume this signal might be related to the sympathetic plexus around the ICA and possibly a small pathway between the skull base and the lymph nodes. We think that our arterial data is not mature and needs more supporting data. Therefore, we removed the arterial data from our study including from the Methods, Results, Discussion, and figures. We are planning a comprehensive study for these arterial findings in the future.

Reviewer #4, Results Comment 5:

There are other recent publications in visualization of meningeal lymphatics which are worth citing - eg Ding et al *Nature Med* 2021; 27:411-418.

Response to Reviewer #4, Results Comment 5:

Thank you so much for the reviewer's suggestion. We are delighted to include this new reference in our paper. We have provided a brief description of these findings in the Introduction. Additionally, we included a new reference about an immunohistochemical study of lymphatic elements in the human brain. In this study, lymphatic marker-positive cells in postmortem human brain samples have been shown in the perineurium and endoneurium of cranial nerves by immunohistochemical technique. The authors suggested that soluble waste may move from the brain parenchyma via perivascular and paravascular routes to the closest subarachnoid space and then travel along the dura mater and/or cranial nerves (Mezey et al, 2021, An immunohistochemical study of lymphatic elements in the human brain. *Proc Natl Acad Sci U S A.*).

Reviewer #4, Results Comment 5:

I did not find the videos at all helpful - these could be dropped.

Response to Reviewer #4, Results Comment 5:

We thank the reviewer for this relevant suggestion. We removed all videos from the main paper and the supplement.

REVIEWER COMMENTS

Reviewer #1 (Remarks to the Author):

The authors have responded forcefully to prior critique. I have no more hesitation regarding this study. The only recommendation is that the authors should acknowledge that fluid efflux from the brain is likely as multifaceted as the lymphatic system in peripheral tissue.

Reviewer #3 (Remarks to the Author):

Thank you for responding to the comments and revising accordingly

Reviewer #4 (Remarks to the Author):

The authors have modified their paper to address many of the reviewers' points.

You repeatedly claim this work to be a 1st visualisation of mlymphatics without iv contrast. This is not true. Absinta et al 2017 (black blood T1) and Cao et al 2020 (heavily T2-weighted FLAIR) have both published on mlymphatics without Gd.

The human sample is a retrospective convenience sample. This information should be presented in the results where the human sample is mentioned. The study was approved by a review board but it is not even clear if patients gave consent for the use of their imaging in this research. This should be clarified.

Line 277 - you say you scanned patients in the prone position - do you mean prone? Most people are scanned supine with MRI due to the shape of the head coils and the fact that the experience is already considered claustrophobic without also having to lie face down. The methods do not mention scanning prone. Lines 166-168 cite gravity and compression of dorsal structure for their non-visualisation implying that patients were supine. Please check and correct.

Figs - Fig 1 - the structure labelled 'dural lymphatic' looks just like ordinary dura - which bit is thought to be the lymphatic? The structures are not consistent with the small channels seen on post contrast imaging. Throughout, the images appear to show dura but not specifically channels. The structures arrowed in Fig 2 just look like dura.

Responses to the Reviewers' Comments:

We greatly appreciate the reviewers' continuing constructive suggestions to improve our manuscript quality. We have addressed the remaining concerns and hope that these revisions will help clear our work for publication.

Reviewer 1 provided a final recommendation that we “*acknowledge that fluid efflux from the brain is likely as multifaceted as the lymphatic system in peripheral tissue.*”

Reviewer 3 noted no additional concerns.

Reviewer 4 had a number of remaining concerns regarding considerations of what aspects of our approach was novel, as well as concerns about patient consent. The reviewer 4 also kindly noted a concern related to patient positioning during imaging and a final concern relating to labels in certain figures.

Reviewer #1, Comment 1:

The authors have responded forcefully to prior critique. I have no more hesitation regarding this study. The only recommendation is that the authors should acknowledge that fluid efflux from the brain is likely as multifaceted as the lymphatic system in peripheral tissue.

Response to Reviewer #1, Comment 1:

We thank the reviewer for their contributions to the improvement of this manuscript. We have addressed that final concern by adding a short passage at the end of the final paragraph of the Discussion.

Reviewer #2, Comment 1:

Thank you for responding to the comments and revising accordingly.

Response to Reviewer #1, Comment 1:

We thank the reviewer for their contributions to the improvement of this manuscript.

Reviewer #4, Comment 1:

You repeatedly claim this work to be a 1st visualisation of lymphatics without iv contrast. This is not true. Absinta et al 2017 (black blood T1) and Cao et al 2020 (heavily T2-weighted FLAIR) have both published on lymphatics without Gd.

Response to Reviewer #4, Comment 1:

We much appreciate the concerns of the reviewer regarding the novelty of our findings in the context of other studies. We have tempered the claim made in the abstract and first sentence of the discussion to provide a more accurate representation of the novelty of our findings.

In light of the reviewers suggestions, we reviewed Absinta et al., 2017 and Cao et al., 2020. In the study by Absinta et al. the researchers used contrast for human MR brain study and marmoset monkey MR. They identified meningeal dural lymphatic structures including T2 FLAIR from human and marmoset subjects, and T1-weighted black blood for humans using two different gadolinium based agents. In their Figure 1, they provide visualization of the meningeal lymphatic structures on contrast enhanced series. Figure 1 second row depicts the right-sided magnified view with red arrows. In the monkey study, they used contrast series for subtraction

from post-gadollinium series and they provided subtracted contrast enhanced series with visualization of the dural lymphatic structures. They mentioned in pre-contrast scans were rigidly registered to post contrast scans then subtraction images were created for anatomical identification of dural lymphatic vessels. They produced 3D images using these outputs. We found no comment or description in this paper of non-contrast FLAIR images for visualization of meningeal structures.

Cao et al., 2020 first performed whole brain MR imaging of dynamic susceptibility contrast changes in the CSF. They tested turbo spin echo (TSE) sequences for evaluation of dynamic signal changes after gadolinium injection along possible lymphatic structures. Using these data, they then estimated Gadolinium concentration in these regions. They used 3D-T2 FLAIR images for localization of anatomical structures useful for measures taken with turbo spin echo dynamic imaging post gadolinium injection. They subsequently fused these images to assess gadolinium concentration. We found no comment or description in this paper of non-contrast FLAIR images for complete visualization of dural lymphatics structures. This paper mainly related to MRI technique and gadolinium quantitative assessment of concentration calculation in possible lymphatic structures.

In light of Reviewer 4's suggestion, we found an additional paper in the literature by Kuo et al., 2018; *Meningeal Lymphatic Vessel Flow Runs Countercurrent to Venous Flow in the Superior Sagittal Sinus of the Human Brain* which depicted lymphatic flow in the parasagittal region using TOF MR angiography techniques. These authors describe visualization of dural lymphatic signals around the sagittal sinus using non-contrast FLAIR images. Our understanding is that this is likely the first report of visualization of MLVs using non-contrast techniques. The authors commented in paragraph 2 of the results that the appearance of the meningeal lymphatic vessels was similar to previously reported MR findings visualized at the corners of the SSS. We did not notice this paper in our preparation of the manuscript, likely due to this paper's focus on TOF MR imaging and direction of lymphatic flow. We have now incorporated this paper into our manuscript.

We agree with the reviewer that different techniques can visualize dural lymphatic structures along the sagittal sinus, whether using IV contrast, no IV contrast, or by intrathecal injection. This finding does not undermine our paper's novelty, as our paper not only evaluated mid-sagittal dural lymphatics but also the entire dorsal and ventral systems and connections of these structures to cervical lymph nodes. We provide a more complete picture of dural lymphatic drainage in live humans compared to previous studies.

Reviewer #4, Comment 2:

The human sample is a retrospective convenience sample. This information should be presented in the results where the human sample is mentioned. The study was approved by a review board but it is not even clear if patients gave consent for the use of their imaging in this research. This should be clarified.

Response to Reviewer #4, Comment 2:

We appreciate the invitation for clarification regarding patient consent in our study. The waiver of informed consent was included within our institution's IRB approval process for this study

given its retrospective nature and de-identified datasets. Mention of this detail has been added to the Methods and Results sections.

Reviewer #4, Comment 3:

Line 277 - you say you scanned patients in the prone position - do you mean prone? Most people are scanned supine with MRI due to the shape of the head coils and the fact that the experience is already considered claustrophobic without also having to lie face down. The methods do not mention scanning prone. Lines 166-168 cite gravity and compression of dorsal structure for their non-visualisation implying that patients were supine. Please check and correct.

Response to Reviewer #4, Comment 3:

We greatly appreciate the reviewer's diligence in catching this error. All patients in the current study were scanned in the supine position, not the prone position. We have corrected this error in the revised manuscript in the line mentioned and added a sentence to the Methods, Patients' MR imaging section, specifying patient posture during scans.

Reviewer #4, Comment 4:

Figs - Fig 1 - the structure labelled 'dural lymphatic' looks just like ordinary dura - which bit is thought to be the lymphatic? The structures are not consistent with the small channels seen on post contrast imaging. Throughout, the images appear to show dura but not specifically channels. The structures arrowed in Fig 2 just look like dura.

Response to Reviewer #4, Comment 4:

We much appreciate the concerns of the reviewer regarding our study's Figures 1 and 2. We agree that there is structural overlap between dural lymphatics and parasagittal dura themselves. Recently, Ringstad and Eide et al., 2020 described the important role of parasagittal dura in CNS lymphatic drainage, clearly demonstrating this concept with MR images taken hours after intrathecal contrast injection (Figures below). Additionally, Absinta *et al.*, 2017 identified lymphatic channels in deceased human histologic specimens. The difference between autopsy findings and *in vivo* volume and diameter of the lymphatic channels in the human brain are not yet known. Generally, in autopsy studies, venous and lymphatic structures collapse, preventing assessment of normal morphology of these structures *post mortem*. All pertinent MR studies in the literature to date have shown lymphatic signature in the parasagittal dura (for example, Absinta et al., 2017; Ringstad & Eide et al., 2020; Kuo, et al., 2018). We provided exhibits below for clarification, including our figure.

Excerpt from Ringstad et al., 2020; **Figure 2**

Excerpt from Absinta et al., 2017 **Figure 1** (left; T2-FLAIR post gadobutrol) and **Figure 2** (right, T1-weighted black blood images post gadobutrol)

Excerpt from Kuo et al., 2018, **Figure 1a**. Non-contrast T2 SPACE FLAIR image.

Image 3D SPACE T2 FLAIR from present study **Figure 1i** (left) and **Figure 2b** (right).

We believe these are sometimes very fine and hard to appreciate, especially in the younger non-atrophic brain. Quality of images of these structures depend on the technique used as well, and, potentially, characteristics unique to certain patient populations. The study by Kuo et al., 2018 depicted very thin 0.5 mm structures in the parasagittal region. Absinta et al., 2017 showed structures of similar dimensions in this location with healthy volunteers. Parasagittal structures identified in the study by Ringstad and Eide et al., 2020 were thicker, more closely resembling findings in the elderly patients of the present study. Though these patterns may be related more directly to underlying disease or advanced age (the patient population was relatively elderly compared to other studies and included individuals with CSF disorders). We believe our findings depicted in Figures 1 and 2 also show parasagittal dural lymphatic signal (not channels). We do not believe any current MR imaging technique can detect meningeal lymphatic canals *per se*, but rather the sum of the lymphatic channels and adjacent lymphatic fluid. Absinta et al., 2017 acquired MR images 31 minutes post-contrast for human study. We suspect that more complete visualization of glymphatic and lymphatic structures requires a greater delay between gadolinium administration and imaging. This hypothesis is supported by the study by Ringstad et al 2020 in which maximum signal change was detected between 6-24 hours post intrathecal injection. Therefore results from Absinta et al., 2017 may represent only the earlier contrast flow patterns detectable on FLAIR images. Even in their study using contrast, visualization of lymphatic structures was better on 2D T1 black blood imaging than 3D FLAIR T2, due to the technical differences. We are currently conducting a study of glymphatic and lymphatic drainage

with IV contrast, and in our personal experience, we only see significant intravascular changes in glymphatic and lymphatic drainage related to gadolinium approximately 30 minutes post contrast administration. Presence of gadolinium in parasagittal structures appears to maximize between 1 hour and 3 hours (unpublished data, manuscript in the preparation).

Below we provide non-contrast 3D T2 FLAIR and post-gadolinium 3D MP RAGE images as further evidence that the parasagittal dural region is more than simple dura. In this figure, stars represent parasagittal dura with signal intensity corresponding to lymphatic macromolecular concentration on FLAIR series. These areas do not enhance on post-gadolinium 3D MP RAGE images, suggesting that these areas are not vascular but also not simple dura, and that their characteristic signal is similar to CSF signal. FLAIR can distinguish lymphatic signal from CSF in this compartment. We hope these explanations can resolve the reviewers' questions and other concerns. We appreciate your detailed and helpful comments, as well as your hard work and commitment in refining this manuscript.

Case 60 from our study group. Right precontrast 3D T2 FLAIR Coronal image and LEFT post contrast T1 MPRAGE at the same level. v:vein, yellow star: parasagittal dural region, and arrows: dura.

REVIEWERS' COMMENTS

Reviewer #4 (Remarks to the Author):

The authors have provided an extensive response to address my remaining concerns. I do not wish to prolong the ping pong match but I do take issue with two of their responses.

the sample - the issue was not just about consent, but about the retrospective sampling: the authors do not address the inevitable limitations of using a retrospective convenience sample; while it may have allowed them to identify mlymphatics, any effort to show apparent associations with patient characteristics in the 81 patient sample, where no details are provided on how they were selected, why they were undergoing MRI in the 1st place, or even the time period, will be biased and not reliable. Line 419 in Discussion and line 597 in Supplement mentions 'we studied only patients with history of seizure disorder' and an 'Epilepsy MR protocol' - were these patients with suspected epilepsy, which was not mentioned as an exclusion? Why not just state this up front? Other overt known diseases are mentioned as excluded but that still leaves wide scope for bias and very limited generalisability of results. They do not mention the total number of potentially usable patient scans from which the 81 were drawn, ie how many were discarded. Nowhere do they even mention the number of males or females, or by age strata. None of these points have been mentioned in the main text methods or discussion of limitations, or in the supplement, but would reasonably be expected in a clinical paper.

lymphatics without IV Gd injection 'for the first time':

- Absinta Fig 1 2nd row left hand image shows mlymphatics on FLAIR precontrast, which then show up more distinctly post Gd:

- Cao et al demonstrate signal change sensitive to lymphatic drainage using T2 with prolonged TE pre Gd (we have used this sequence ourselves to visualise mlymphatics in patients; however the thin small vessels are easier to visualise after Gd on FLAIR) - while they do not make a big issue out of the precontrast images, the information is nonetheless present in their paper;

- Kuo does not use iv Gd and demonstrates the same dural lymphatic vessels.

other minor points:

Abstract line 51: '...in the ventral nerves in the human brain...' should be '..ventral nerves of the human...' or even 'exiting' not 'in'. the nerves are outside the brain when they acquire their dural coverings.

the intra-rater reliability - no disrespect to the neuroradiologist who has obviously done a large amount of work for this paper, but it is not mentioned if the repeated measures were blind to each other.

lymphatic drainage around lower cranial nerves - Introduction line 69-70, is overstated and please make it clear that you refer to humans in vivo since there is a reasonable literature on dural drainage around major arteries and cranial nerves exiting the skull eg in sheep and other animals and some from humans pm - it is a credible and reliable literature and should not be overlooked.

Human PM lower skull drainage described in:

- Weller, R. O., Djuanda, E., Yow, H. Y. & Carare, R. O. Lymphatic drainage of the brain and the pathophysiology of neurological disease. *Acta Neuropathol* 117, 1-14, doi:10.1007/s00401-008-0457-0 (2009).

- Ha, S.-K., Nair, G., Absinta, M., Luciano, N. & Reich, D. Magnetic resonance imaging and histopathological visualization of human dural lymphatic vessels. *Bio-Protocol* 8, doi:10.21769/BioProtoc.2819 (2018).

other relevant references on the topic:

Louveau, A. et al. Structural and functional features of central nervous system lymphatic vessels. *Nature* 523, 337-341, doi:10.1038/nature14432 (2015).

Eide, P., Vatnehol, S., Emblem, K. & Ringstad, G. Magnetic resonance imaging provides evidence of lymphatic drainage from human brain to cervical lymph nodes. *Scientific reports* 8, 7194, doi:10.1038/s41598-018-25666-4 (2018).

Responses to the Reviewers' Comments, Fourth Revision

We greatly appreciate the reviewer's continuing constructive suggestions to improve our manuscript quality. We have addressed the remaining concerns and hope that these revisions will help clear our work for publication. Below is a brief itemized list of changes and responses pertinent to reviewers' most recent concerns:

Reviewer #4, Comment 1:

The sample - the issue was not just about consent, but about the retrospective sampling: the authors do not address the inevitable limitations of using a retrospective convenience sample; while it may have allowed them to identify lymphatics, any effort to show apparent associations with patient characteristics in the 81 patient sample, where no details are provided on how they were selected, why they were undergoing MRI in the 1st place, or even the time period, will be biased and not reliable. Line 419 in Discussion and line 597 in Supplement mentions 'we studied only patients with history of seizure disorder' and an 'Epilepsy MR protocol' - were these patients with suspected epilepsy, which was not mentioned as an exclusion? Why not just state this up front? Other overt known diseases are mentioned as excluded but that still leaves wide scope for bias and very limited generalisability of results. They do not mention the total number of potentially usable patient scans from which the 81 were drawn, i.e. how many were discarded. Nowhere do they even mention the number of males or females, or by age strata. None of these points have been mentioned in the main text methods or discussion of limitations, or in the supplement, but would reasonably be expected in a clinical paper.

Response to Reviewer #4, Comment 1:

Thank you for addressing this concern. We agree that the selection criteria should be clear for a retrospective study. Indeed, we already provided a supplementary figure (previously *Supplementary Figure 1*) in past drafts of the manuscript. In this figure, we show our inclusion and exclusion criteria and the impact each criterion had on patient candidacy for the study. As this figure depicts, we started with 236 patients imaged with our technique between August 2018 and September 2019 due to clinical history of epilepsy or suspicion of epilepsy. We first eliminated 121 patients from the study due to contrast use before the time of imaging. After this, we excluded an additional 17 patients due to image quality concern (e.g. motion artifacts). Finally, we excluded another 17 patients out of concern for abnormal anatomy, including patients with a history of surgery, status epilepticus, mass lesion, significant atrophy, acute diffusion restriction on DWI, or focal parenchymal lesion. Patients undergoing sedation during imaging were also excluded, as sedation may increase or decrease glymphatic flow and lymphatic flow as shown in animal studies. While these details were not easily accessible in previous drafts of the manuscript within the supplementary files, we understand that these details are important for readers and reviewers of the paper. We have therefore made our main *Figure 2*, including sex information and age information, what was previously *Supplementary Figure 1*.

We also provided additional information regarding limitations of the study at the end of the *Discussion* of the previous draft. We provided additional information in the *Methods* section regarding our selection technique and inclusion/exclusion criteria and why we chose only patients with a history of epilepsy or suspicion of epilepsy without any history of surgery, status epilepticus, mass lesion, significant atrophy, acute diffusion restriction on DWI, or focal parenchymal lesion.

The patients with these conditions were excluded because these can cause anatomical distortion or focal lesion that may confound interpretation of lymphatic structures. Additionally, status epilepticus or acute seizure might change lymphatic flow and enhance visualization of the lymphatic structures. We are aware that this is a retrospective study only within patients with a history of epilepsy or suspicion of epilepsy. These are not necessarily healthy, normal subjects. This represents a limitation in our study. We already discussed this limitation in the *Discussion*, limitations section, of the previous draft (lines 419 and 420 of previous draft). We provide more elaboration in the present draft regarding this limitation.

We also thank you for your concerns regarding transparency of data relating to study subjects. In the previous draft, *Results* section, *Study Human Subjects*, lines 148 and 149, we stated “A total of 81 human subjects (45 females and 36 males) with a mean age of 41.7 (SD 20.4, range 15-80) years were included in this retrospective study.” We agree with the reviewer that these details are not easily accessible in the main text for readers, therefore we included this information in *Figure 2* of this latest revision.

Reviewer #4, Comment 2:

Lymphatics without IV Gd injection 'for the first time':

- Absinta Fig 1 2nd row left hand image shows lymphatics on FLAIR precontrast, which then show up more distinctly post Gd:

- Cao et al demonstrate signal change sensitive to lymphatic drainage using T2 with prolonged TE pre Gd (we have used this sequence ourselves to visualise lymphatics in patients; however the thin small vessels are easier to visualise after Gd on FLAIR) - while they do not make a big issue out of the precontrast images, the information is nonetheless present in their paper;

- Kuo does not use iv Gd and demonstrates the same dural lymphatic vessels.

Response to Reviewer #4, Comment 2:

We agree with Reviewer 4; in this revision we removed all definitively worded statements of novelty of our findings. We removed all usages of the terms “novel,” “first,” and similar words and phrases making these suggestions. In light of the reviewer’s suggestions, we referenced the papers referred to by the Reviewer in the previous revision.

Reviewer #4, Comment 3:

Abstract line 51: '...in the ventral nerves in the human brain...' should be '..ventral nerves of the human...' or even 'exiting' not 'in'. the nerves are outside the brain when they acquire their dural coverings."

Response to Reviewer #4, Comment 3:

Thank you for the input. Per your suggestions, we have instead opted for use of the phrase “ventral regions/parts in the human brain” in the Abstract as well as the rest of the manuscript.

Reviewer #4, Comment 4:

The intra-rater reliability - no disrespect to the neuroradiologist who has obviously done a large amount of work for this paper, but it is not mentioned if the repeated measures were blind to each other.

Response to Reviewer #4, Comment 4:

We agree with reviewer 4. All repeated measurements were done blind to each other. We have now provided information for intra-rater variability in the *Results* section.

Reviewer #4, Comment 5:

Lymphatic drainage around lower cranial nerves - Introduction line 69-70, is overstated and please make it clear that you refer to humans in vivo since there is a reasonable literature on dural drainage around major arteries and cranial nerves exiting the skull eg in sheep and other animals and some from humans pm - it is a credible and reliable literature and should not be overlooked.

Human PM lower skull drainage described in:

- Weller, R. O., Djuanda, E., Yow, H. Y. & Carare, R. O. Lymphatic drainage of the brain and the pathophysiology of neurological disease. Acta Neuropathol 117, 1-14, doi:10.1007/s00401-008-0457-0 (2009).

- Ha, S.-K., Nair, G., Absinta, M., Luciano, N. & Reich, D. Magnetic resonance imaging and histopathological visualization of human dural lymphatic vessels. Bio-Protocol 8, doi:10.21769/BioProtoc.2819 (2018).

other relevant references on the topic:

Louveau, A. et al. Structural and functional features of central nervous system lymphatic vessels. Nature 523, 337-341, doi:10.1038/nature14432 (2015).

Eide, P., Vatnehol, S., Emblem, K. & Ringstad, G. Magnetic resonance imaging provides evidence of glymphatic drainage from human brain to cervical lymph nodes. Scientific reports 8, 7194, doi:10.1038/s41598-018-25666-4 (2018).

Response to Reviewer #4, Comment 5:

In light of reviewer 4's suggestion, we re-wrote the *Introduction* line 69-70 and included references to the studies noted by the reviewer.